



# The consistency between observations (TCCON, surface measurements and satellites) and $CO_2$ models in reproducing global $CO_2$ growth rate

Lev D. Labzovskii[1], Samuel Takele Kenea[1], Jinwon Kim[1], Haeyoung Lee[2], Shanlan Li[1], Young-Hwa Byun[1], Tae-Young Goo[1], Young-Suk Oh[1]

[1]Climate Research Division, National Institute of Meteorological Sciences, Seogwipo, Jeju, 63568, Republic of Korea
[b]Environmental Meteorology Research Division, National Institute of Meteorological Sciences, Seogwipo, Jeju, 63568, Republic of Korea

*Correspondence to*: Samuel Takele Kenea (samueltake@yahoo.com)

**Abstract.** Atmospheric $CO_2$ growth is the primary driver of the global warming and the rate of this growth is a valuable indicator of the interannual changes in carbon cycle. Despite atmospheric $CO_2$ growth rate had been considered as the well-known quantity, the latest findings indicated that $CO_2$ models can considerably disagree in reproducing this rate. This study is aimed to advance our knowledge about temporal and spatial variations of annual $CO_2$ growth rate (AGR) by using $CO_2$ observations from the Total Column Observing Network (TCCON), $CO_2$ simulations from Carbon Tracker (CT) and Copernicus Atmospheric Monitoring System (CAMS) models being compared with the previously-reported global references of AGR from Global Carbon Budget (GCB) and satellite observations (SAT) for 2004-2019 years. TCCON and the $CO_2$ models revealed temporal AGR variations ($AGR_{TCCON}$ = 1.71 – 3.35 ppm, $AGR_{CT}$ = 1.64 – 3.15 ppm, $AGR_{CAMS}$ = 1.66 – 3.13 ppm) of very similar magnitude to the global $CO_2$ growth references ($AGR_{GCB}$ = 1.59 – 3.23 ppm, $AGR_{SAT}$ = 1.55 – 2.92 ppm). However, $AGR_{TCCON}$ estimates agree well with the references only during the 2010s (correlation coefficient, r = 0.68 vs GCB and r = 0.75 vs SAT) as the TCCON observational coverage has been substantially expanded since 2009. Moreover, $AGR_{TCCON}$ reasonably agrees (r = 0.67) with the strength of El-Nino Southern Oscillations (ENSO) in the 2010s. The highest atmospheric $CO_2$ growth (2015-2016) driven by the very strong El-Nino was accurately reproduced by TCCON which provided AGR of 2015-2016 years (3.29 ± 0.98 ppm) in very close agreement to the $AGR_{SAT}$ reference (3.23 ± 0.50 ppm). We further validated AGR simulations (CT and CAMS) versus the newly-acquired $AGR_{TCCON}$ (as point-location reference) for every TCCON site and found low agreement between the models and TCCON (r < 0.50) only at 3 out of 20 stations. This minor caveat has not affected the accuracy of global AGR simulations as they showed high agreement with SAT (r ≈ 0.76 - 0.78) and GCB (r ≈ 0.72 – 0.78) and reasonable agreement with TCCON (r = 0.65) global-scale references. The spatial correlation between CT and CAMS in simulating AGR (applied for every 3° x 2° grid cell) is perfect (r = 0.99) for the modeling period (2004-2016). Similarly, land-wise intercomparison between CAMS and CT simulations of AGR yielded in perfect correlation for most MODIS land classes (median of land-dependent r > 0.98). From spatial perspective, the highest AGR estimates (> 20% from the median) were observed in the regions of intense fossil fuel combustion (East Asia) or biomass burning (Amazon, Central Africa). Lack of ideal correlation and small disagreement



between CT and CAMS (< 3.9 % difference between medians of global AGR estimates) are likely driven by the slight spatial disagreement between CT and CAMS in the aforementioned regions. To validate this statement, a sensitivity experiment is needed where in $CO_2$ inverse model, alongside with the current setup of a priori biomass burning fluxes, an alternative setup is assembled (multiple independent estimates of burned area and fire-dependent emission factors for various type of tropical fires can be used). In overall, our study showed that the current estimates of global atmospheric growth rate of $CO_2$ are consistent across a wide range of the different data sources and strengthening of carbon observational infrastructure (like covering more developing countries with ground-based $CO_2$ observations and providing more satellite $CO_2$ observations from cloudy and hazy regions) should improve the accuracy of $CO_2$ growth rate estimates on both local and global scales.

## 1. Introduction

During the last 50 years, the world has been witnessing a permanent growth in the atmospheric $CO_2$ concentration (Betts et al., 2016; Keenan et al., 2016), a principal driver of the global warming (Lacis et al, 2010; Stips et al., 2016). Atmospheric $CO_2$ growth rate (GR) is steadily sustained in the modern times given constantly increasing fossil fuel $CO_2$ emissions ($FFCO_2$) and weakening of carbon sinks due to global warming (Canadell et al., 2007; Schneising et al., 2014; Friedlingstein et al., 2015). As the stability and the precision of direct observations are high (0.09 ppm), the temporal variability of GR in the entire atmosphere is known with high confidence and the latest estimates are uploaded to the Global Carbon Budget (GCB) at regular basis (Le Quere et al., 2018). The estimates of global-scale GR are fairly robust since they are derived from stable surface atmospheric measurement of $CO_2$ mole fraction taken at the multiple stations worldwide (Dlugokencky and Tans 2018). Despite that, it has been recently shown than different models can disagree in reproducing global $CO_2$ growth and this disagreement constrains accurate partitioning between various carbon fluxes (terrestrial and ocean) in the models (Gaubert et al., 2019). The gaps in the knowledge about GR can stem from poorly understood spatio-temporal characteristics of GR in the atmosphere. The temporal changes of GR are prominently manifested at yearly scales and above all indicate the climate-driven changes in terrestrial fluxes (Alden et al., 2010). Hence, annual growth rate of atmospheric $CO_2$ (AGR) naturally varies and these variations are fundamentally controlled by El-Nino Southern Oscillation (ENSO), $FFCO_2$ and the dynamics of terrestrial carbon sink (Buchwitz et al., 2007; Keenan et al., 2016; Kim et al., 2016; Ekwurzel et al., 2017; Buchwitz et al., 2018). The role of $FFCO_2$ in AGR variations is fairly simple as the emissions load extra carbon to the atmosphere, thus increasing the atmospheric $CO_2$ (Ekwurtzel et al., 2017). In turn, ENSO is a more intricate driver since it indirectly affects AGR by altering the temperature-water regime of ecosystems (Zeng et al., 2005; Wang et al., 2013; Kim et al., 2017) or by causing fires and vegetation disturbances (Jones and Cox 2005; Liu et al., 2017; Chylek et al., 2018) that ultimately alter terrestrial carbon sink. At global scales, AGR exhibits strong temporal variations (~ 2.0 ppm $yr^{-1}$) whereas





the anomalies beyond this rate are either caused by the outstanding El-Nino events (like in 2015/2016 according to Betts et al., 2018) or by the powerful volcanic eruptions (like Pinatubo eruption of 1992 according to Frolicher et al., 2013).

From a spatial perspective (at point locations or within constrained spatial domain), AGR may differ from global-scale growth. This difference stems not only from the time lag between the moment when $CO_2$ is released to the atmosphere (or absorbed by land) and the moment when $CO_2$ is well-mixed throughout the entire atmosphere. It also stems from the unique response of local carbon pools to extreme meteorological conditions, forest fires and to deforestation process (House et al., 2002). Supposedly, AGR spatial variations are mainly controlled by anomalies of temperature (Rafelski et al., 2009; Schneising et al., 2014) and precipitation (Poulter et al., 2014). Furthermore, the meteorological anomalies can trigger sound changes in terrestrial water storage which is a strong driver of spatial AGR changes per se (Jung et al., 2017; Piao et al., 2019). The roles of ecosystem and their vegetation are therefore pivotal for AGR spatial heterogeneity and the strongest drivers originate in tropical (Cox et al., 2013; Wang et al., 2013; Kim et al., 2017; Rodenbeck et al., 2018) and semiarid regions (Ahlstrom et al., 2015). Despite the regions that drive global-scale AGR were explicitly determined in the aforesaid studies, the AGR spatial variability is still poorly understood. The evidences of AGR spatial variability at various scales are occasionally reported by using ground-based observations (Fang et al., 2014), spaceborne measurements (Schneising et al., 2014; Liang et al., 2017; Buchwitz et al., 2018) and $CO_2$ inverse models (Cheng et al., 2013; Nayak et al., 2014; Labzovskii et al., 2019). From the satellite data, we learned about the large-scale spatial variations of AGR (~ 0.5 ppm yr$^{-1}$) that emerge at latitude bands comparable to the geographically extensive climate zones (Buchwitz et al., 2018). In turn, at local scales, the information about AGR variability is available only for the limited number of stations (Fang et al., 2014). In an ideal position, a global observational network could provide regular $CO_2$ measurements, so GR at monthly and annual scales would be calculated for nearly each location in the world (being included by the measurement footprint of a certain station). Unfortunately, the carbon research infrastructure is technically far from the inception of all-encompassing $CO_2$ observations. On the ground, both in-situ (Ciais et al., 2014) and remote sensing atmospheric observations (Wunch et al., 2014) of $CO_2$ are substantially limited in the majority of developing countries. The modern satellites provide precise $CO_2$ measurements that are still plagued by low signal-to-noise ratio in the cloudy and hazy regions (Kim et al., 2016; O'Dell et al., 2018). The inadequate understanding of land carbon sinks (Peylin et al., 2013), the lack of information about location of carbon sources and sinks (Peters et al., 2017), the weakness in attributing the physical drivers of anomalous AGR by both land carbon (Piao et al., 2019) and earth system models (Keppel-Aleks et al., 2014); all impose additional modelling-related limitations for regional AGR analysis.

The main aim of this study is therefore to advance our knowledge about temporal and spatial variations of global $CO_2$ atmospheric growth. To pursue this aim, we intercompare versatile $CO_2$ datasets including newly-analyzed data (ground-based remote sensing and $CO_2$ inverse modelling) and the existing AGR global-scale references in 2004-2019 years. Two new $CO_2$ data sources include (1) globally-aggregated TCCON (Total Carbon Column Observing Network) observations



(Wunch et al., 2014) and (2) a set of two $CO_2$ inverse models including CarbonTracker (Peters et al. 2007) and Copernicus
Atmospheric Measurements System (Chevalier et al., 2013) that are referred to CT and CAMS, respectively. The existing
global AGR references are Global Carbon Budget (Le Quere et al., 2018) and the satellite estimates from the previous
impactful study about $CO_2$ growth (Buchwitz et al., 2018). To advance our knowledge about AGR, this study pursues three
objectives. At first, to (a) assess the consistency of global AGR estimates (from aggregated TCCON data and from $CO_2$
modelling) with the existing observational references. This is required not only to understand the consistency of AGR
estimates across various datasets, but also to understand the TCCON suitability as a reference for validating point-scale
simulations of AGR by the $CO_2$ models. Once we approve the $CO_2$ models' ability to reproduce AGR spatial variations
(using comparison with TCCON), the rest objectives are to (b) estimate spatio-temporal inconsistencies in AGR simulations
by two $CO_2$ inverse models (CAMS and CT) and to (c) determine which regions (i.e. which ecosystems) can drive these
inconsistencies. The novelty of our study can be summarized by three points. At first, we deploy TCCON and $CO_2$ inverse
models to complement existing knowledge of AGR. At second, we explicitly investigate spatial variability of AGR using
$CO_2$ models. At third, we analyze whether $CO_2$-associated factors (such as ENSO and vegetation type) can affect the overall
agreement in the AGR intercomparison between two different models.

This manuscript is organized as follows. Sect. 2 presents methodology. Sect. 3 contains results. Sect.s 4 and 5 represent
discussion and conclusions respectively.


## 2. Data and methodology

### 2.1 Main datasets

This Sect. describes the data we use in this study. We present the main tools for retrieving $CO_2$ atmospheric
concentration, and also describe the additional datasets (used for determining strength of ENSO events and for checking the
land cover type) that support our analysis.

### 2.1.1 TCCON Network

TCCON (the Total Carbon Column Observing Network) provides continuous measurements of column-averaged dry-air
mole fractions of $CO_2$ around the globe ($XCO_2$). The $XCO_2$ estimates are retrieved using the ratio of column abundance of
$CO_2$ to column abundance of O2. Observations of $XCO_2$ are widely recognized as less sensitive to variations in surface
pressure (and atmospheric vapor as well) so they are perfectly suitable for $CO_2$ investigation at different geographic and





climate conditions. The TCCON observation principle is based on ground-based Fourier transform spectrometry (FTS) that offers high spectral (0.02 cm-1) and temporal resolution (~ 90 s) spectra by pointing on the sun in the near-infrared spectrum. The use of TCCON observations of $XCO_2$ is advantageous as ground-based FTS are highly robust and allow calculating

$XCO_2$ parameters at ~ 0.8 ppm (~ 0.25%) accuracy after calibration (Wunch et al., 2011). It is to be noted that $XCO_2$ products from TCCON had been calibrated using collocated aircraft observations deployed with the reference instrumentation from World Meteorological Organization onboard (Messerschmidt et al., 2011). Moreover, TCCON provides quality-assured $XCO_2$ data (Wunch et al., 2014) since the observations are acquired and processed using the same set of standardized tools and methods (including instrumentation, data acquisition routines, processing software and

calibration procedures). Today, TCCON excels with a quasi-global observational cover (despite many gaps still exist) and 27 operational sites are dispersed around the globe. A reader may familiarize with all stations used in this work from Fig. 1 (the full list of stations and the respective acronyms are shown in the supplementary material; see Table S.1.1). Our work addresses to the TCCON data from February 2004 to April 2019 and we use the data from both operational sites and the sites where the measurement activity has been halted. The entire study period for AGR examination is chosen to be consistent

with the previous prominent study about AGR (Buchwitz et al., 2018). The accurate method for reproducing local-scale AGR (at point location) is certainly required to be accounted as the "fine-scale reference" for evaluating AGR simulations from the $CO_2$ models. TCCON observations can fit the requirements to become such a reference since TCCON data had been numerously used for validation purposes against global-scale tools such as satellites (Morino et al., 2011; Liang et al., 2017; Wunch et al., 2017) and $CO_2$ models (Kulawik et al., 2016). Most importantly, TCCON showed great efficiency in

reproducing AGR at a single location where the estimates have been compared with the CT results (Yuan et al., 2019). TCCON data were obtained from the online depository (http://tccon.ornl.gov) with the latest access on November 2019.





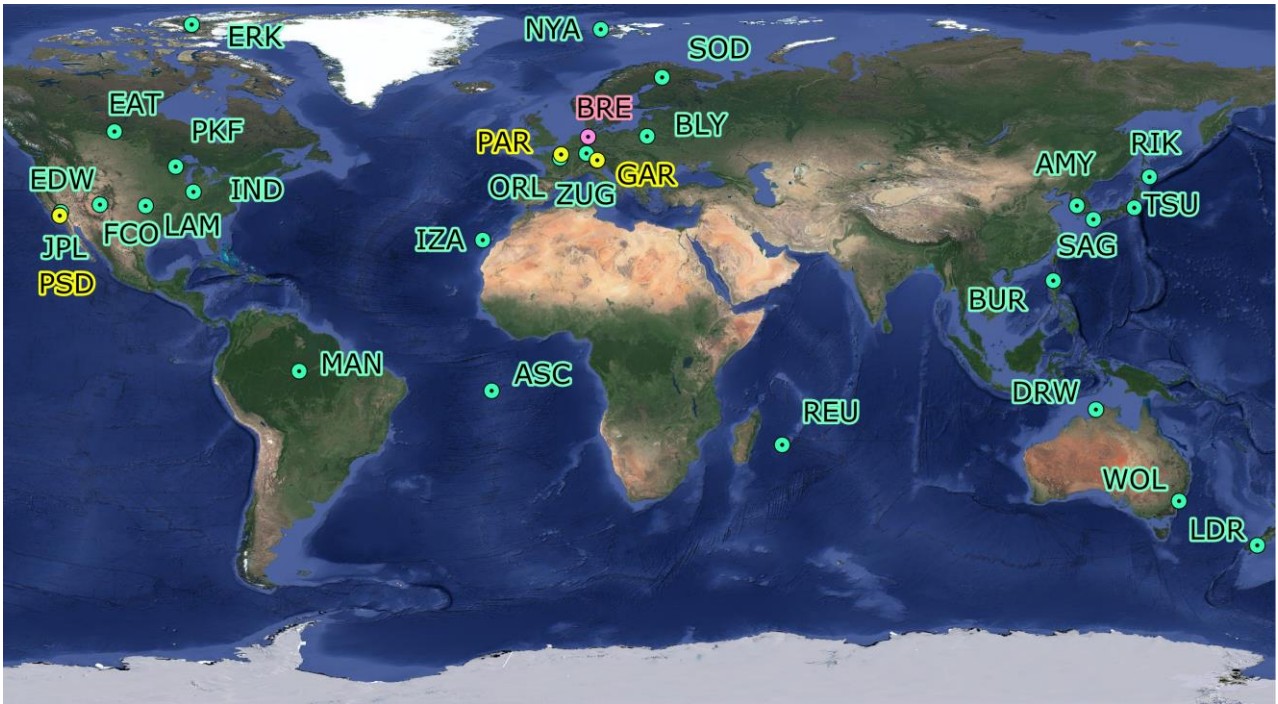

**Figure 1. Geographical distribution of TCCON sites. Colors are shown for better distinguishing neighboring measurement sites.**
**Description of acronyms used for each measurement site and detailed maps of the regions congested with several TCCON stations**
**are provided in the supplementary material. The satellite map provided by Google maps is embedded using QuickMapServices**
**plugin**

**2.1.2 Copernicus Atmospheric Monitoring Service**

Copernicus Atmospheric Monitoring System (CAMS) is based on data assimilation of $CO_2$ observations and being developed by the Institute of Pierre Simon Laplace (LSCE). For CAMS, atmospheric observations of $CO_2$ are compiled via several networks of near-surface observations including the NOAA Earth System Research Laboratory archive, the World Data Centre for Greenhouse Gases, the Integrated Carbon Observation System-Atmospheric Thematic Center and Réseau Atmosphérique de Mesure. For simulating the atmospheric transport of $CO_2$, CAMS utilizes the LMDZ transport model (Locatelli et al., 2015). The mass fluxes are determined from a full General Circulation Model controlled by the ECMWF (European Centre for Medium-Range Weather Forecasts) winds (Chevallier et al., 2005; Chevallier et al., 2010; Chevallier, 2013). Prior values of the fluxes are provided to the climatological land-atmosphere fluxes at a 3-hour resolution from the ORCHIDEE model (ORganizing Carbon and Hydrology In Dynamic EcosystEms), gridded annual anthropogenic emissions (combined by Carbon Dioxide Information Analysis Center i.e. CDIAC and Emission Database for Global Atmospheric Research; i.e. EDGAR). Ocean fluxes and biomass burning module are defined based on the sea-air fluxes and GFED





(Global Fire Emission Database), respectively (before 2014, in 2015 fire module uses Global Fire Assimilation System). In this study, we use CAMS v18.2 version of PYVAR atmospheric inversion with an initial spatial resolution of ~ 3.7° x 1.9° (longitude-latitude) for the entire atmospheric column (integrated over 39 vertical levels). We note that we regridded CAMS results to 3° x 2° longitude-latitude grid (to be compatible with CarbonTracker) and this process did not cause serious error

propagation to the estimates due to similarities in the models' spatial resolutions. We downloaded CAMS $XCO_2$ results from the online depository of ECMWF based on data request from November 2019 (https://apps.ecmwf.int/datasets/data/cams-ghg-inversions).

### 2.1.3 CarbonTracker

CarbonTracker (CT) global version (Peters et al., 2007) is also based on the data assimilation of $CO_2$ observations. CT was originally developed by the National Oceanic and Atmospheric Administration (NOAA) for better understanding the processes governing $CO_2$ uptake and release at the Earth surface in high temporal and spatial resolution. CT exploits TM5 (Transport Model 5) offline atmospheric transport model (Krol et al., 2005) forced by the ERA-Interim meteorological fields from the ECMWF (the European Centre for Medium-Range Weather Forecasts). CT offers $CO_2$ simulations with global

cover of 3° x 2° degree (longitude - latitude) resolution. Like other $CO_2$ inverse models, CT uses various carbon fluxes (oceanic and land) to optimize with existing $CO_2$ observations. To define the a priori $CO_2$ fluxes from various carbon sources, CTA utilizes modules including the land-atmosphere carbon exchange (from Carnegie-Ames Stanford Approach model), wildfires (Global Fire Emissions Database; i.e. GFED), fossil fuel emissions (from "Miller" dataset and Open-Data Inventory for Anthropogenic Carbon Dioxide, i.e. ODIAC) and oceans (Ocean Inversion Fluxes). Kalman filter is used as an

optimization method and the latest CT version (2017) assimilates hourly-averaged $CO_2$ concentration from 254 observations sites worldwide. This study uses $CO_2$ simulations from CarbonTracker 2017 version (denoted simply as CT). We emphasize that CT offers $CO_2$ concentration at 25 height levels. We use pressure-weighted integrated columnar concentration of $CO_2$ (to be consistent with CAMS integrated estimates of $CO_2$) that was derived according to methodology shown in Zhao et al., (2019). The details about this calculation are provided in the supplementary material (see Sect. S3). The CT datasets are

downloaded     from     NOAA     website     being     accessed     on     November     2019 (ftp://aftp.cmdl.noaa.gov/products/carbontracker/co2/molefractions/).

### 2.1.4 Datasets with global CO2 growth rate

For global $CO_2$ growth data, we exploited the two types of estimates. On the one part, we use Global Carbon Budget-2018

dataset (Le Quere et al., 2018). GCB introduces the comprehensive investigation of the all four components of carbon cycle



(including FFCO$_2$, land-use emissions, oceanic sink, land sink and the atmospheric CO$_2$ growth rate) and being updated at a regular basis. Obviously, among all components we used atmospheric GR that had been taken from GCB-2018 (Le Quere et al., 2018). Estimates of AGR from GCB-2018 were directly provided by atmospheric CO$_2$ concentration measurements with the uncertainties of ~ 0.2 Gt/C yr$^{-1}$ (i.e. ~ 0.4 ppm) according to the data provider (Dlugokencky and Tans, 2018). We used only the data from 2004 onwards, with the latest available GR estimates on the time of manuscript preparation. GCB dataset was downloaded via the website of the journal of Earth System Data (https://www.earth-syst-sci-data.net/10/2141/2018/) with the latest access on November 2019. Another reference for large-scale AGR estimates (global scales, hemispheric scales and tropical latitude bands) are taken from the satellite data analysis conducted by Buchwitz et al., (2018) study (see Table A1 of the corresponding manuscript). Their satellite-based estimates (SAT hereafter) were acquired via combining spectrometric CO$_2$ observations from SCIAMACHY (SCanning Imaging Absorption Spectrometer for Atmospheric CHartographY) and GOSAT (Greenhouse gas Observing SATellite) satellites for 2003-2016 period.

### 2.1.5 Ancillary datasets

We also use CO$_2$-related data including ENSO indices and land cover classification data. El-Nino (warm) and La-Nina (cold) ENSO events were quantified using Oceanic Nino Index (ONI) obtained from the online platform (https://ggweather.com/enso/oni.htm) of Golden Gate Weather Services (Null et al., 2013). For comparing ENSO events with atmospheric growth rates projected for every month, we used the quantitative descriptors of ENSO (i.e. used running 3-month mean ONI values). For comparing with AGR, we exploited quantitative metrics labeled "ENSO type". According to this quantitative typology, ENSO events can be annually classified as weak, moderate, strong and very strong El-Nino (labeled as 1, 2, 3, 4 respectively in this work) and as weak, moderate, strong and very strong La-Nina (labeled as -1, -2 , -3, -4 respectively in this work).

To understand which land type each grid cell (that contain CO$_2$ data) belongs to, we acquired Land Cover data from Terra and Aqua of MODIS (Moderate Resolution Imaging Spectroradiometer) dataset (V3, 2013). This dataset is originally titled "Global Land Cover" and had been developed by the National Mapping Organizations (Tateishi et al., 2011, 2014; Kobayashi et al., 2017). Global land cover dataset offers high spatial resolution product (500 m). Land cover product includes 20 types of surfaces (the full list of the land types can be found in the supplementary Table S.2.1) whereas the land type largely depends on the vegetation features. The land cover product has been obtained in the form of raster files (4 datasets with 90 x 180 tiles by latitude and longitude respectively) from the online file depository (https://globalmaps.github.io/glcnmo.html) with the latest access on November 2019.



### 2.2 Methodology for CO2 atmospheric growth rate calculation

In the subsection devoted to the methodology of our research, we describe the principles of GR calculation.

### 2.2.1 Calculation of CO2 growth rate at monthly and annual scales

The recent comprehensive review about interannual variations of carbon cycle has emphasized that AGR may considerably vary depending on the methodology of calculation (Piao et al., 2019). It implies that there is no conventional methodology for calculating AGR, and we suggest that the method should be selected according to the latest findings in the research literature. In this regard, we refer to the latest outcomes from the GR research and use the methodology of Buchwitz et al. (2018) whereas their method originates from the commonly-accepted GR calculation approach (Thoning et al., 1989).

According to this method, prior to calculating AGR we need to derive the annual growth of $CO_2$ sampled for each month (MGR hereafter) during the year. In this way, the MGR concept of this study corresponds to what was referred as "annual $CO_2$ growth rate of specific month" (Wang et al., 2014) or as "monthly sampled $XCO_2$ annual growth rate" (Buchwitz et al., 2018) in the previous studies. Eq. (1) shows that MGR simply represents the difference between $XCO_2$ of each month during a year ($i$) and the corresponding same month of the previous year ($i$-12). Therefore, MGR can be calculated if $CO_2$ estimates

are available for a certain month in two consecutive years.

$$MGR(XCO_2) = XCO_{2(i)} - XCO_{2(i-12)}$$

(1)

At the next step, AGR is calculated by taking all-month (12 months in perfect case) median of MGR during the same year. The corresponding uncertainties are taken from the standard deviation of AGR (stemming from MGR variability within the same year).



## 3. Results

### 3.1 Analyzing TCCON ability to provide point-based AGR

### 3.1.1 Aggregated global-scale AGR from TCCON versus global-scale references (Global Carbon Budget and Satellites)

In the first instance, we evaluate the ability of TCCON observations to reproduce accurate GR using the chosen methodology. To this end, we collected all available observations from the TCCON sites from February 2004 onwards and
calculated monthly medians of $XCO_2$ for each station. For the 2004-2019 period, we initially had 30 stations with available observations. At each station, the number of available $XCO_2$ observations (further used for global-scale MGR calculation) vary from 2 to 60 per month. We omitted all MGRs that had been calculated using less than 2 observations during a month due to lack of confidence in such averages. In general, the perfect temporal observational cover was not expected given regular forfeiture of FTS observations due to cloudy conditions (Wunch et al., 2011). Thus, the total number of MGR values
at the TCCON station greatly varies depending on the geographic location and consequently most MGR estimates were collected from the stations in North America, Australia and Europe. The total final number of stations that meet the aforesaid requirements is 20. We collected all TCCON-based MGR estimates ($MGR_{TCCON}$) from these stations and aggregated them together (the bottom part of Fig. 2, panel a) for further calculation of global-wide AGR using TCCON measurements (the upper part of Fig. 2, panel a). Then, we compare the resulted TCCON-based AGR ($AGR_{TCCON}$) estimates with the referenced
global data (GCB and SAT) and also analyze the agreement of all types of AGR with ENSO metrics (given the strong fundamental linkage between AGR and ENSO).

At first sight, MGR estimates reasonably reflect one-year growth of $CO_2$ over a certain location (thin bars of Fig. 2, top panel). Given constantly increasing $CO_2$ in the entire atmosphere around the globe, it is reasonable that ~98% of all $MGR_{TCCON}$ are positive connoting location-independent $CO_2$ growth. Then, we calculate global-scale AGRs in 2006-2019
period using the aforementioned $MGR_{TCCON}$ (2004 and 2005 years are missing due to absence of sufficient FTS observations). In 2006-2019 years, $AGR_{TCCON}$ ranges from 1.71 ppm (2009) to 3.35 ppm (2008). There is no large difference in the magnitude of AGR variations between the global references ($AGR_{SAT}$ = 1.59 - 3.23 ppm, $AGR_{GCB}$ = 1.55 - 2.95 ppm). However, $AGR_{TCCON}$ poorly agrees with both GCB and SAT estimates as correlation coefficient (r) in TCCON-to-GCB agreement is 0.32 and in TCCON-to-SAT agreement it is 0.28 (see top panel of Fig. 2). Notably, the global references (SAT
and GCB) exhibit much stronger agreement with each other (r = 0.88) in AGR reproduction. Nevertheless, we suppose that the poor agreement of TCCON with the references is not an indication of inadequate accuracy of AGR retrieved by TCCON observations. The disagreement most likely stems from the uneven presence of TCCON observations. More specifically, the total number of MGRs (see bottom panel of Fig. 2) has been steadily increasing over a study period from MGRs based on 2 TCCON stations (2007) to MGRs based on 21 stations (2018). Due to this, for the latest decade (2010-2019) $AGR_{TCCON}$ has
higher agreement with the referenced AGRs (r = 0.61 for TCCON-SAT, r = 0.49 for TCCON-GCB). It is unlikely that





shortening of the analyzed period is the driver of the improved agreement, since similar curtailment of analysis to 2004-2009 period does not affect the agreement of global-scale $AGR_{TCCON}$ with the global references (r < 0.30 for both SAT and GCB).

ENSO strength and ONI indices are illustrated on Fig. 2 (bottom panel) right below GR plot for better comprehension of qualitative agreement between GR and ENSO parameters. From Fig. 2, we evidence the ENSO role in forming wave-shape fluctuations of MGR (from TCCON) throughout the study period. One can suspect that seasonality (i.e. vegetation cycles) is responsible for wave-shape fluctuations of MGR but it is scarcely probable since seasonal medians of MGR are nearly equal. Namely, medians of MGR are 2.39 ppm, 2.29 ppm, 2.36 ppm and 2.36 ppm for winter, spring, summer and autumn respectively (see Fig. S.4.1 in the supplementary material for further details). Alas, it is challenging to approve the quantitative agreement between MGR and ENSO due to difficulties in matching three-month period ONI indices with 1-month based MGRs. MGR response on short-scale ENSO condition can substantially vary by time so the ultimate agreement will depend on the chosen "lag" between ENSO event and MGR (Kim et al., 2016). Meanwhile, the agreement between the annual strength of ENSO (right axis of Fig. 1, panel b) with the referenced AGR is easily discernible and seems reasonable (r = 0.67 for $AGR_{GCB}$ and r = 0.64 for $AGR_{SAT}$). At the same time, the agreement between ENSO strength and $AGR_{TCCON}$ is yet very low (r = 0.27). Once again, when only observation-abundant decade (2010-2019) is analyzed, $AGR_{TCCON}$ exhibits as reasonable agreement with ENSO (r = 0.67) as the global references do. Moreover, the improvement in AGR-to-ENSO agreement is not only associated to data availability but also to the enhanced role of ENSO in global $CO_2$ growth. It is known, that the current role of ENSO (post-2010 period) in forming interannual fluctuations of GR is pivotal (94%) compared to the earlier periods of time (63% in post-2003 period) (Buchwitz et al., 2018). This effect is clearly manifested in our analyzed data by the presence of stronger AGR-ENSO agreement in 2010-2016 period (r = 0.76 for $AGR_{GCB}$ and r = 0.66 for $AGR_{SAT}$) compared to 2004-2009 period (r = 0.57 for $AGR_{GCB}$ and r = 0.56 for $AGR_{SAT}$). Besides 15-year pattern, we noticed that the aggregated TCCON data can provide global-scale AGR that is sensitive to strong ENSO events. More specifically, during the record El-Nino event (Betts et al., 2016; Liu et al., 2017; Paek et al., 2017; Betts et al., 2018; Buchwitz et al., 2018) TCCON observations yield in very high AGR (3.29 ppm ± 0.98 ppm) similar to the highest $AGR_{SAT}$ (3.23 ppm ± 0.50 ppm) or $AGR_{GCB}$ (2.85 ppm ± 0.09 ppm) estimates during that year (see year 2015/2016 on Fig. 2, panel b). We underscore that AGR-2016 from TCCON is not the highest during the study period of TCCON observations due to year 2010 when the accuracy of TCCON estimates of AGR (3.35 ppm) has been seemingly affected by high uncertainty (5.02 ppm) given scarcity of stations used for AGR calculation during that year.









**Figure 2. Panel a: Comparison of AGRTCCON (gray line) versus AGRGCB (brown line) and AGRSAT (black line). Vertical thin**
**bars stand for MGR (that are used for ultimate AGR calculation). AGRs are plotted on x-axis versus eighth month (August) of**
**each year; second months of the year (February) are shown alongside since the first observation from TCCON is available starting**
**from February 2004. Error bars of AGRTCCON (gray vertical dashed line) are defined based on standard deviation of AGR**
**across all stations using monthly-mean-based weighting (see details in supplementary material in Table S.1.2 and the**
**corresponding description). Errorbars for AGRSAT and AGRGCB are omitted since their corresponding uncertainties are lower**
**and temporally less variable than the uncertainties of AGRTCCON (0.27 - 0.50 ppm for SAT and 0.4 ppm for GCB). Panel b: ONI**
**indices (vertical bars) for various three-months periods (see bar colors where JJA stands for June-July-August, JAS for July-**
**August-September etc.) during the year where periods starting from winter months are shown with green tones, from spring with**
**gray tones, from summer with golden tones, from autumn with red tones. Annual ENSO strength is plotted (bold line) with the**
**information about ENSO year-to-year transition (red line - transition towards warmer anomaly, blue line - transition towards**
**colder anomaly, black line - no interannual change).**

### 3.1.2 Evaluating sensitivity of AGR estimates from TCCON depending on data abundance on the monthly scales

The agreement between AGR$_{TCCON}$ with the references (and with ENSO as well) during the entire period of study is low and
the concerns about the suitability of TCCON data for retrieving AGR data can be risen. Due to this, a brief sensitivity
analysis is given below. We examine how the threshold for calculating MGRs (based on one-time TCCON observations)
may affect the error spread of AGR by TCCON estimates. Namely, spread expresses the range of AGR variability just due to
MGR type that used in the AGR calculation input. To this end, we tested seven sub-monthly thresholds for calculating MGR
data (2, 3, 5, 10, 15, 20 and 30 observations) as shown on Fig. 3. Tightening the threshold from 2 to 3 points (or to 5 points)
would not result in any significant loss of MGR data. However, switching 2-point-based threshold to more stringent options
would result in significant loss of data with 8-22% of annual MGR forfeited at 10 points (compared to 2 points availability),
8-35% at 15 points and 13-39% at 20 points. Bottom panel of Fig. 3 shows that shifting to 30-points threshold results in
critical loss of MGR data (38-100% of data loss per year compared to 2 points availability). At the same time, the magnitude
of AGR estimates (depending on MGR abundance) would not be soundly affected during most years (see lines at bottom
panel of Fig. 3). The standard deviation across various AGRs (calculated using different thresholds we mentioned above)
would range between 0.03 ppm (2019) to 0.42 ppm (2006). In terms of potential error spread, AGR experiences from 1.5%
(2019) to 21.0% (2006) spread depending on the MGR input (Fig. 3, top panel). It is reasonably to suggest that the error
spread could be entirely driven by the number of available TCCON stations (i.e. less stations, higher error rate). However, it
is unlikely the case since the correlation between AGR error spread and number of total available MGR (stemming from
number of stations) is low (r = -0.42) and comparable to the strength of correlation between error spread and ENSO (r = -
335   0.47).



**Figure 3. Bottom panel: AGR global estimates depending on the threshold of TCCON observations used for MGR calculation per month (denoted as AGR-i format where i stands for the minimum threshold) are shown by bold colored lines. Bars represent the total number of available MGR used for AGR calculation per year (denoted as CNT-i format). Top panel: error spread across hypothetical AGR global estimates (gray bars) stemming from the range of used threshold (for MGR calculation) shown by percent.**

In Table 1, we present the final evaluation of this subsection where TCCON agreement with references and with ENSO (as shown in Fig. 2) is tested based on the various observation thresholds that we discussed above. We found that these agreements negligibly change over the range of thresholds from 2 to 15 ($r < 0.50$ in all cases) while dramatic improvement in the agreement is discerned at 20-point threshold ($r = 0.61$ for TCCON-SAT, $r = 0.52$ for both TCCON-GCB and TCCON-



ENSO comparison). Shifting to more stringent thresholds (as 30 points) leads to a sharp deterioration of agreement rates (likely due to critical loss of MGR availability incompatible with global-scale AGR quantification). We underline that the current analysis approves the sanity of TCCON data (at annual scales). Since we are interested to keep the most data-

abundant set of AGR$_{TCCON}$ estimates (for evaluating $CO_2$ models) and there is no large difference between "2 points" and more stringent thresholds until "20 points", we use the most data-abundant threshold of "2 points". We emphasize that future studies should use as many points as possible where 20 sub-monthly TCCON observations (to provide robust MGR average) is preferable minimum threshold for further calculation of the aggregated global-scale AGR$_{TCCON}$.

**Table 1. Sensitivity analysis of AGR due to MGR threshold (using TCCON data). MGR Threshold denotes the minimum number of sub-monthly observations that should be available for calculating MGR. Three last columns provide correlation coefficients for TCCON-SAT, TCCON-GCB and TCCON-ENSO agreement when global AGR is calculated for the period of study. The optimum threshold is highlighted by the bold font in the table.**

| MGR Threshold | r (TCCON-SAT) | r (TCCON-GCB) | r (TCCON-ENSO) |
|---|---|---|---|
| 2 | 0.27 | 0.32 | 0.27 |
| 3 | 0.28 | 0.33 | 0.26 |
| 5 | 0.27 | 0.32 | 0.27 |
| 10 | 0.30 | 0.23 | 0.23 |
| 15 | 0.29 | 0.18 | 0.31 |
| **20** | **0.61** | **0.52** | **0.52** |
| 30 | 0.23 | 0.27 | 0.29 |


**3.2 Evaluating simulated MGRs (CT and CAMS) compared to TCCON references**

In this paragraph, we compare all MGRs simulated by CAMS and CT versus TCCON estimates of MGR. This step is required to understand how well the $CO_2$ models can simulate MGR at point locations (and therefore spatial variability of MGR as well). To this end, we compiled all data-abundant TCCON stations (where at least 18 of MGRs during > 1.5 year of

observations should be available). The result is shown on Fig. 4 where correlation coefficients from TCCON-CT (blue) and TCCON-CAMS (red) comparisons for the 20 TCCON stations are given. The modeled MGRs agree well with the MGR$_{TCCON}$ references. For CT, correlation coefficient ranges from 0.12 to 0.87 (median = 0.66) and for CAMS from 0.11 to




0.89 (median = 0.64). We identified very high agreement between CT and TCCON (r ≥ 0.80) at 3 stations, high agreement at 4 stations (0.70 ≤ r ≤ 0.79), reasonable agreement at 5 stations (0.60 ≤ r ≤ 0.69) and weak agreement at 4 stations (0.50 ≤ r ≤

0.59). For CAMS the results are similar as 4, 3, 5 and 4 stations exhibited very high, high, reasonable and weak agreements with TCCON references (with the same classification of agreement ranges as shown above). Since not all stations exhibit acceptable agreement rates in TCCON-to-model comparison (r ≥ 0.50), it is prudent to analyze what drives the deteriorated agreement at some stations.

According to CT and CAMS, the lowest agreement (in reproducing MGR) with TCCON (r < 0.50) is found for 3 stations

including Tsukuba (r = 0.12 for CT and 0.11 for CAMS), Pasadena (r = 0.34 for CT and 0.38 for CAMS) and Ascension (r = 0.39 for CT and 0.44 for CAMS). It is unlikely that the deteriorated agreement is entirely associated with regional vegetation patterns since there is no correlation between MGR and latitude in either of the models. Such vegetation proxy may seem plain but the latitude of station provides a simple and realistic indicator to vegetation activity when atmospheric $CO_2$ is analyzed (Graven et al., 2013). Since these stations are located in the different ecosystem regions (see Table S.1.1. in

supplementary material), the low agreement is also unlikely related to ecosystem flux prescription of the $CO_2$ models. Closer examination of TCCON-to-model comparison hints that each station has some unique features of disagreement. For instance, at Tsukuba station (where the lowest agreement has been found), the largest discrepancy between the modeled and the observed MGRs is evidenced at the peak of land carbon sink activity in northern hemisphere (May-to-August). In particular, both strong overestimation of MGR (d = MGR$_{TCCON}$ - MGR$_{MODEL}$ < - 2.5 ppm) and underestimation (d = MGR$_{TCCON}$ -

MGR$_{MODEL}$ > 3.0 ppm) of MGR by models can be marked at such periods. For the Ascension station, there is no season-dependent agreement patterns. This can be explained by the overwhelming exposure of the station to oceanic carbon fluxes compared to diminished role of land fluxes. Most likely, Ascension is a small island, which cannot be resolved by the model resolutions as land and is rather treated as ocean. For the Pasadena station, due to abundant observations, it is relatively easy to identify the driver of disagreement. MGR$_{TCCON}$ at Pasadena exhibits such distinct seasonal cycle so MGR amplitude can

be discerned. The MGR amplitude can be understood as a difference between the highest and the lowest value of MGR during the year (that is seemingly driven by $CO_2$ amplitude as well). Simulated bottom-to-peak annual amplitude of MGR is nearly always lower than the observed MGR amplitude from TCCON. Namely, CT substantially underestimates the MGR$_{TCCON}$ amplitude by 2.61 (78%), 1.67 (45%) and 0.20 ppm (12%) in 2014, 2015 and 2016 years, respectively. CAMS exhibits even stronger underestimation of MGR$_{TCCON}$ amplitude by 2.86 (86%), 1.66 (44%) and 0.67 ppm (40%) ppm in

2014, 2015 and 2016, respectively. The underestimation of simulated $CO_2$ amplitude compared to TCCON observations is a common phenomenon in the northern hemisphere that most likely stems from underestimation of the carbon uptake by the models in high-latitudes (Peng et al., 2015). A reader can familiarize with the aforementioned station-wise MGR temporal dynamics we discussed above in the supplementary material (Fig. S.4.2 in the supplementary material).





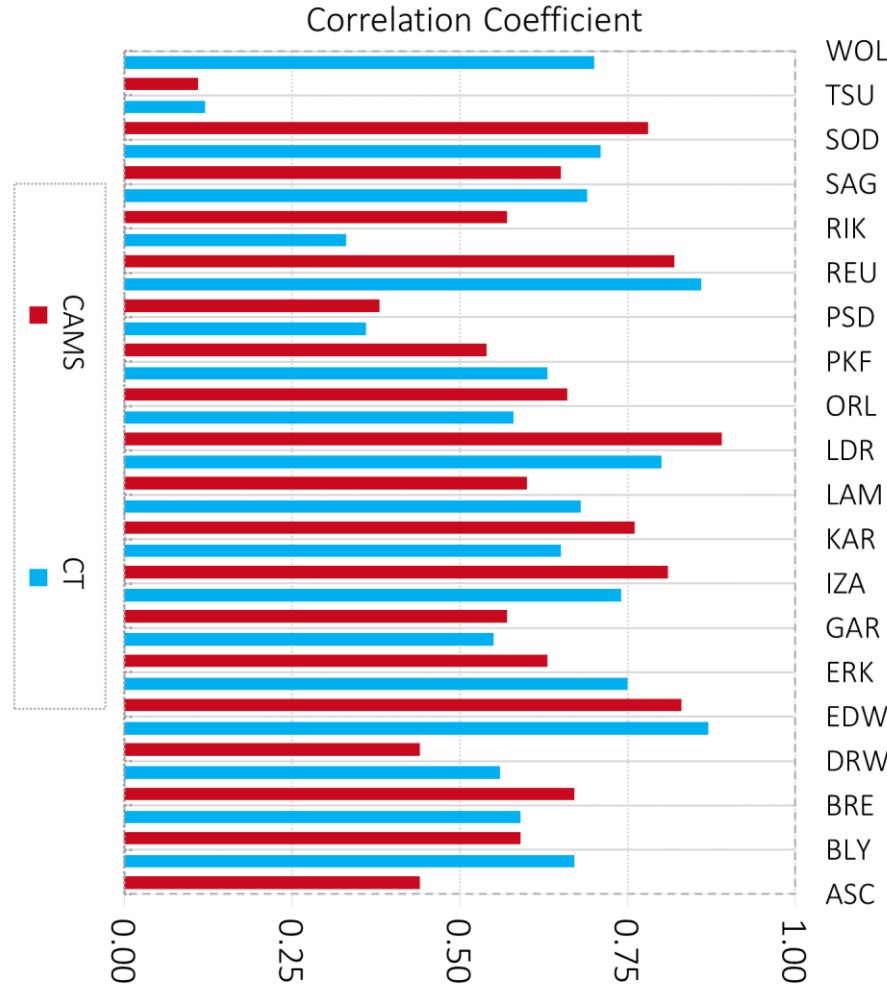

**Figure 4.** Correlation coefficients between all available MGRs for various TCCON stations and MGRs modeled by CAMS (red) and CT (blue). Simulations are provided for the same location and given at monthly temporal resolution.

### 3.3 Evaluating simulated global-scale AGRs (CT and CAMS) compared to TCCON, Global Carbon Budget and satellite global-scale references

We further examine the ability of the $CO_2$ models to reproduce global-scale AGR. We analyze simulated AGR (from CT and CAMS) versus global-scale AGR from the ground-based reference (TCCON) and the global references from the previous studies (SAT and GCB). Such evaluation of the models is a good functional quality test that may reveal hidden caveats of the $CO_2$ modelling process at the spatial scale of interest (Chevallier et al., 2011). The evaluation encompasses broad spatial





scales including global cover, mid-latitudes of both northern hemisphere (a range of 30 - 60º N latitude bands denoted as NH

hereafter) and southern hemisphere (defined within 60 - 30º S range and denoted as SH hereafter) as well as tropical band (30º S - 30º N). Global scale analysis (top-left panel of Fig. 5) shows that AGR varies from 1.66 to 3.13 ppm (CAMS) and from 1.64 to 3.15 ppm (CT). Simulated AGRs are generally consistent with the observational estimates according to $AGR_{SAT}$ (median = 2.21 ppm for this period), $AGR_{GCB}$ (2.09 ppm) and $AGR_{TCCON}$ (2.08 ppm) as well. For the entire period of study (including 2017, 2018 and 2019 years that are missing in the models), $AGR_{TCCON}$ is higher than $AGR_{SAT}$, $AGR_{GCB}$ and

modeled AGR (both CT and CAMS) on 54%, 46% and 62%, respectively. It is an interesting finding since the previous studies had shown that TCCON estimates may underestimate global $CO_2$ atmospheric growth (Chevallier et al., 2011). From statistical perspective, we discern high agreement between the modeled AGRs and both $AGR_{SAT}$ (r = 0.78 for CAMS and r = 0.76 for CT) and $AGR_{GCB}$ (r = 0.74 for CAMS and r = 0.72 for CT) at global scales. The agreement between modeled AGR and $AGR_{TCCON}$ is lower but yet reasonable (r = 0.65 for both models). We also notice the change in model-to-reference

agreement patterns at the finer spatial scales. More specifically, in NH, the models exhibit perfect correlation with $AGR_{TCCON}$ (r = 0.90 for CAMS and r = 0.91 for CT). However, model-to-SAT agreement is weakened down to the reasonable rates of agreement in NH (r = 0 .61 for both models). In SH, we observe very similar but stronger agreement patterns in both TCCON-to-model (r = 0.96 for both models) and TCCON-to-SAT comparisons (r = 0.70 for CAMS and r = 0.74 for CT). Likewise, in tropical regions, the TCCON-to-model agreement yields in 0.95 correlation coefficient for both

models and TCCON-to-SAT agreement varies depending on the model (r = 0.78 for CAMS and r = 0.73 for CT).



**Figure 5. Evaluating ability of CO₂ inverse models (CT in red and CAMS in blue) to reproduce global CO₂ growth at annual scales (AGR). Modeled results are compared with SAT (black), GCB (yellow) and TCCON (gray) references. Top panels represent analysis for global scales and mid-latitude (30º - 60º N) northern hemisphere (i.e. NH) scales. Bottom panels represent analysis for mid-latitude (60º S - 30º S) southern hemisphere (i.e. SH) and tropical (30º S - 30º N) scales.**

## 3.4 Analysis of AGR agreement between CT and CAMS. Spatial perspective.

We have identified the high agreement between the modeled and referenced (i.e. observational-based) AGR estimates in the previous section. Since the CO₂ inverse models may provide spatial variability of AGR far beyond the observational cover of TCCON network or the satellites, we use this opportunity to compare models' ability to reproduce spatially heterogeneous AGR. To this end, we compare the modeled AGRs with each other (CT vs CAMS) at different dimensions. We note that the





difference between AGR estimates provided by two different models can be associated to the mismatch of the observation sites used for assimilation, to the transport-model-driven errors and to the time lag between the biosphere activity and $CO_2$ growth since end-year fluxes may affect AGR of the next year (van der Laan-Luijkx et al., 2017). At first, we analyze the

finest spatial dimension of the agreement by testing grid cell correlation. Grid cell correlation is applied for every single grid cell that represents minimum spatial domain of the model (i.e. 3º x 2º) for the entire modelling period (2004-2016). Figure 6 shows the spatial analysis with medians of AGRs (CT and CAMS), as well as the differences and a grid cell correlation between CAMS and CT medians of AGR. Median $AGR_{CT}$ in 2004-2016 period (calculated for each grid cell) spatially ranges from 1.77 to 2.35 ppm depending on the location (median of 2.01 ± 0.07 ppm). $AGR_{CAMS}$ grid cell-resolution

estimates in 2004-2016 period are very similar and range from 1.82 to 2.40 ppm (median of 2.03 ± 0.08 ppm). These AGR estimates agree well to the widely-acknowledged average global AGR of ~ 2.00 ppm reported by most previous studies. We further try to detect long-term spatial anomalies from the AGR distribution. All the geographic regions with 20%-increased AGR (arbitrary threshold) by CT seemingly represent the areas where the strong biomass burning events frequently occur (minor part of Amazon, Central Africa) or the areas with active fossil fuel combustion (East Asia). The same regions are

visible from $AGR_{CAMS}$ distribution. Despite minor discrepancies, the models almost perfectly agree with each other. Left-bottom panel of Fig. 6 shows that the mean difference (d) between $AGR_{CT}$ and $AGR_{CAMS}$ estimates (in 2004-2016 period) spatially varies from -0.06 to 0.16 ppm range (median d = 0.01 ppm ± 0.02). Likewise, the correlation between CT and CAMS (right-bottom panel of Fig. 6) in reproducing spatial variability of AGR is very high and ranges from 0.88 to 1.00 (r = 0.99 in median that indicates perfect agreement). From spatial perspective, the agreement is very high whereas the small

geographic variability of the agreement is yet visible. Correlation and difference analyses (bottom panels of Fig. 6) show that nearly perfect agreement between CAMS and CT is found over the Oceans, Australia and over all land surfaces with sparse vegetation. The only areas with slightly deteriorated agreement between the models are seen over the regions with abundant and productive vegetation and in the areas where biomass burning (and combustion) is very intense.



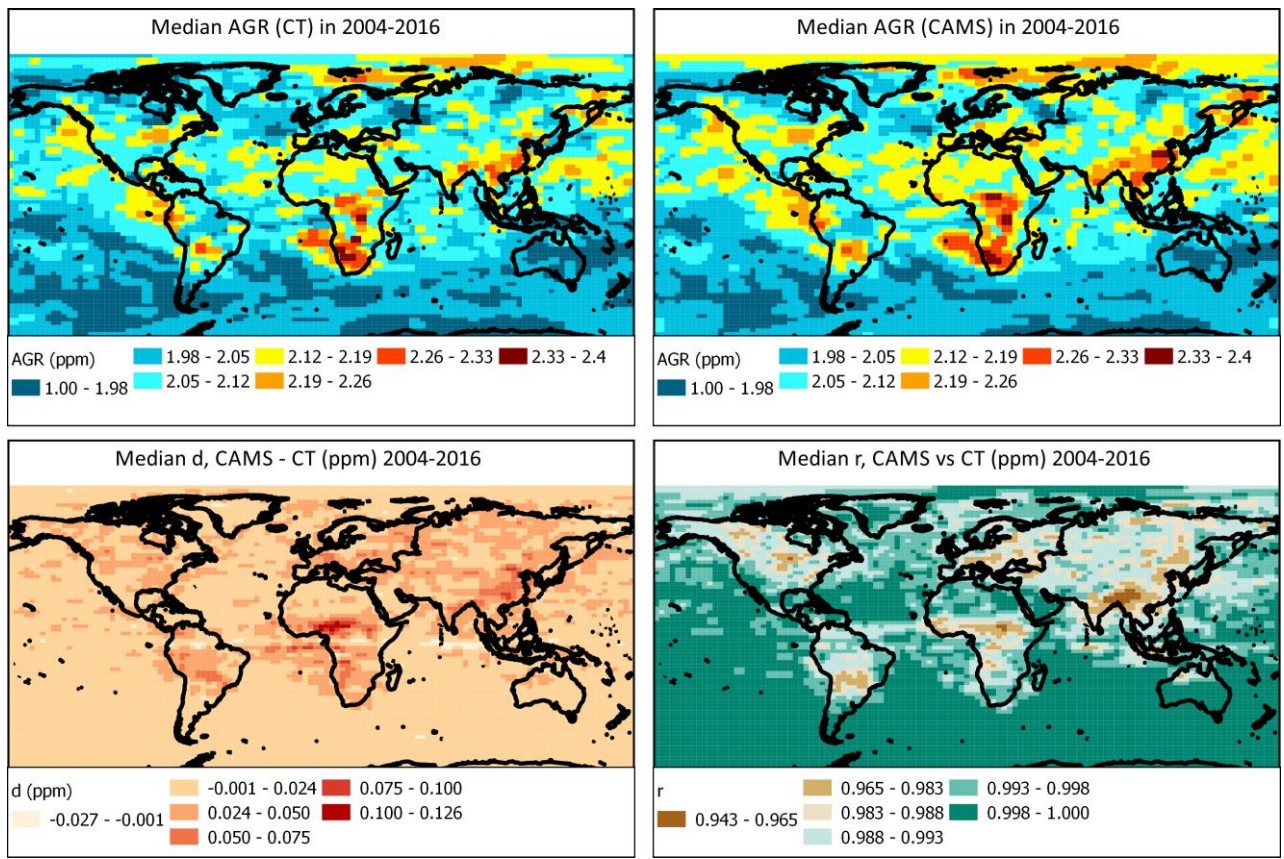

**Figure 6. Top panel: median estimates of AGR from the models (CT on the left side and CAMS on the right side) in 2004-2016 period. Bottom panel: left subplot represents difference (d) between AGR estimates (CT and CAMS) in 2004-2016 period (median for the entire period is shown), right subplot represents grid cell correlation (r) between two modeled AGRs (CT and CAMS). All estimates are provided with 3º x 2º resolution.**

Based on the findings shown above, we suggest that there is a relationship between inter-model agreement rate (in reproducing AGR) and the vegetation type. To check this premise, we analyze AGR correlations ($r_{AGR}$) and differences ($d_{AGR}$) between CAMS and CT in details. Thus, we classify all the grid cells that contain AGR information according to land cover classification (using MODIS data). We take into account that land cover analysis is not a main objective of this study and we maximally simplify this analysis by assuming that every model grid cell (3º x 2º) corresponds to one specific land type (detailed description is given in supplementary Sect. S.2). Therefore, $r_{AGR}$ and $d_{AGR}$ are calculated for each land type (among 20 types) as shown in Fig. 7. We found out that the agreement between the models in reproducing AGR is almost perfect across all land types. Namely, mean $r_{AGR}$ ranges from 0.98 (paddy field) to 1.00 (snow surfaces, water bodies and bare areas). Since $r_{AGR}$ negligibly varies within each land cover type (σ of $r_{AGR}$ < 0.01 within any land cover type), first and third quartile of $r_{AGR}$ both vary in narrow range of 0.98 - 1.00 (that is is also very similar to median of $r_{AGR}$). It should be



noted that the analysis of model differences indicates to sources of model disagreements in a better way. General agreement between models is very high that is evident from the following statistics. Median $d_{AGR}$ ranges from 0.00 ppm (snow surfaces) to 0.04 (urban areas, paddy fields, wetland and herbaceous tree/shrub cover). All the latter land cover types l exhibit the highest variability of model agreement within their domains as σ of $d_{AGR}$ for these land covers is the highest (0.04 ppm for urban areas and 0.03 ppm for the rest aforementioned land covers and mangrove as well). The first quartile (25%) are equal

to 0.03 ppm (urban), 0.03 ppm (herbaceous tree/shrub), 0.02 ppm (wetland) and 0.02 ppm (mangrove) for these land cover types. The third quartile (75%) are 0.05, 0.06, 0.04, 0.05 ppm for the same respective land cover types. The highest agreement is found over all land cover types with no vegetation (snow surfaces and water bodies with median $d_{AGR} = 0.01 \pm 0.01$ ppm). Such vegetation-dependent agreement between models is explainable given the pivotal role of biosphere in the modelling of interannual signal of $CO_2$ atmospheric growth.






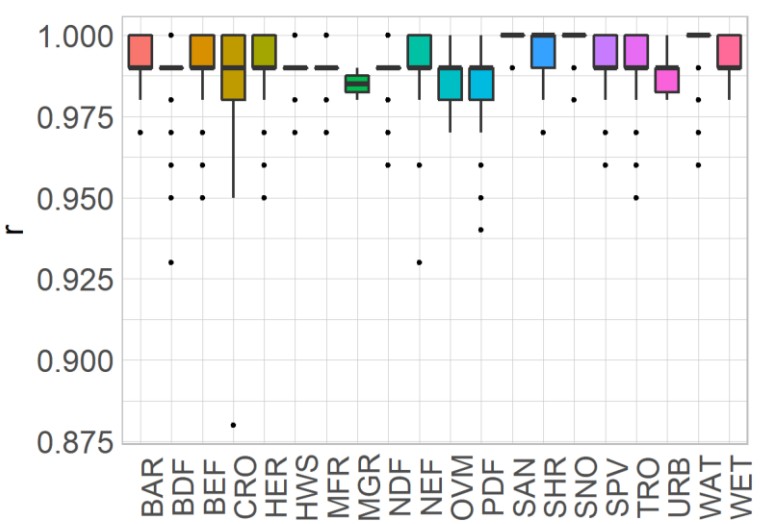

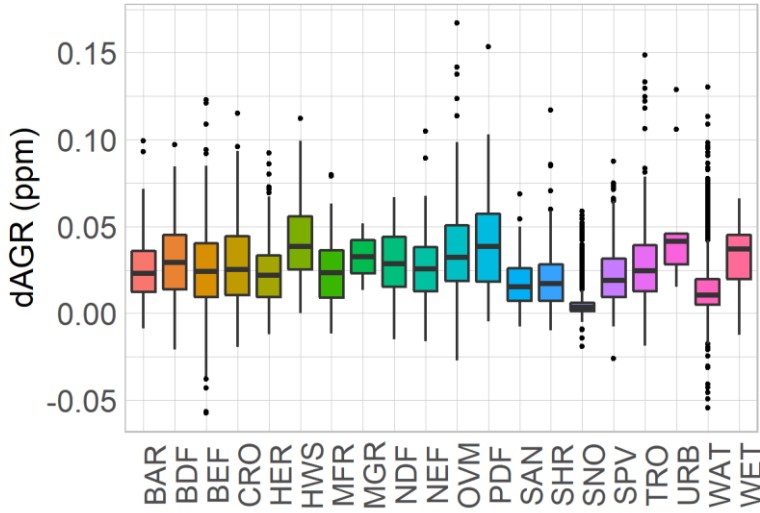

**Figure 7. Top panel: boxplot with correlation coefficients AGR_CAMS and AGR_CT for different land types. Bottom panel: boxplot with differences between AGR_CAMS and AGR_CT for different land types. Colors denote different land types.**



Despite nearly ideal spatial agreement in reproducing AGR between CAMS and CT, we have noticed some model differences in several regions arguably related to areas of strong biomass burning and intense fossil fuel combustion. Regional analysis shown in Table 2 approves our suggestion about weakening of model AGR agreement in such regions. Despite perfect spatial agreement (r = 0.99 - 1.00), the largest model differences in simulating AGR are reported over East Asia (d = 0.04 ppm) that is the strongest FFCO$_2$ source region in the world (Le Quere et al., 2018) and over the Middle-West

Africa (d = 0.04 ppm) where biomass events are frequently reported using MODIS data by GFED (Randerson et al., 2012). If frequent biomass burning is a driver of a discrepancy, the absence of South-East Asia is rather surprising. The latter region is also prone to frequent wildfires and largely located in the tropical zone where the highest discrepancies in biosphere fluxes from various CO$_2$ inversions was recently reported (Kenea et al., 2020). Oppositely, the smallest and the most negligible differences are found over the regions with no vegetation (d = 0.00 ppm over Antarctica) or where ocean fluxes are

predominant (d = 0.01 ppm over oceans and Australia-Oceania as well). We note that the differences between the regions with the best and the worst agreements is minimal given nearly perfect agreement of the models at spatial dimension analysis. Despite this fact, persistent but maximal discrepancy between the models (such as in Mid-West Africa) can highlight important contrasts between the models' setups. Such contrasts can arise due to the differences in the model's biosphere a priori estimates of by discrepancies in atmospheric transport where CO$_2$ level is high. Since this analysis encompasses the

finest spatial domain of a single grid cell as a subject of analysis, one can state that the important discrepancies between modeled AGRs at global scales (due to these spatial discrepancies) could be overlooked. To this end, we also quantify temporal (annual) agreement (as median correlation coefficient for the entire broad spatial domain for every single year). Hence, temporal correlation can be retrieved for every year and embraces the broad spatial domain (such as global, NH, SH or tropical scales). Temporal (annual) correlation in AGR estimates between CAMS and CT is also perfect. We omitted the

plot about temporal agreement since median annual AGR estimates by the models were shown on Fig. 5 and the temporal correlation insignificantly varies from year to year. More specifically, temporal correlation is always perfect between CAMS and CT. It ranges from 0.97 to 0.99 at global scales, from 0.94 to 0.99 in NH, from 0.96 to 0.99 in SH and from 0.96 to 0.99 in tropics.

**Table 2. Calculating median differences (CAMS - CT) and correlation coefficients (CAMS vs CT) of AGR for various regions worldwide. Spatial boundaries of these regions can be found in the supplementary material (Fig. S.2.2).**

| Type | Difference | Correlation |
|---|---|---|
| Antarctica | 0.04 | 0.99 |
| Australia-Oceania | 0.05 | 0.97 |





| | | |
|---|---|---|
| Oceans | 0.08 | 0.96 |
| Central America | 0.10 | 0.94 |
| North Africa | 0.10 | 0.94 |
| Middle East | 0.14 | 0.93 |
| North America | 0.13 | 0.93 |
| Europe | 0.17 | 0.92 |
| South Asia | 0.13 | 0.92 |
| South Africa | 0.11 | 0.92 |
| South-East Asia | 0.12 | 0.91 |
| Central Asia | 0.16 | 0.91 |
| Asian Russia | 0.18 | 0.90 |
| South America | 0.14 | 0.90 |
| Middle-West Africa | 0.14 | 0.89 |
| East Asia | 0.20 | 0.88 |

## 4. Discussion

To advance our knowledge about $CO_2$ atmospheric growth, this study profoundly investigated the temporal and spatial variations of $CO_2$ atmospheric annual growth rate (AGR) using ground-based observations of $XCO_2$ (TCCON), the $CO_2$ inverse models (CT and CAMS) alongside with the global-scale references of AGR from Global Carbon Budget (GCB) and from satellite data (SAT) for 2004-2019 years.

At global scales, the newly-used datasets reveal temporal AGR variations ($AGR_{TCCON}$ = 1.71 – 3.35 ppm, $AGR_{CT}$ = 1.64
– 3.15 ppm, $AGR_{CAMS}$ = 1.66 – 3.13 ppm) of similar magnitude to the referenced estimates ($AGR_{GCB}$ = 1.59 – 3.23 ppm, $AGR_{SAT}$ = 1.55 – 2.92 ppm). $AGR_{TCCON}$ shows low agreement with GCB (r = 0.32) and SAT (r = 0.28) in the period of 2004-2016 years. The deteriorated agreement of TCCON with the references is not associated to the accuracy of TCCON observations (annual error propagation due to sub-annual input of AGR calculation varies in 1.5 - 21.0 % range). It is related to the temporal irregularity of data availability by the TCCON network. In the data-abundant decade (the 2010s), the
agreements between $AGR_{TCCON}$ with the references are dramatically improved (r = 0.68 vs GCB, r = 0.75 vs SAT). Despite





spatial constraints of the TCCON network, globally-aggregated FTS observations are reasonably sensitive to ENSO-driven changes of AGR. In particular, the agreement between ENSO strength and $AGR_{TCCON}$ is reasonable (r = 0.67) in the 2010s when ENSO events played an exclusive role (94%) in forming AGR variability. The strongest atmospheric $CO_2$ growth (2015-2016) caused by the very strong El-Nino event was accurately reproduced by the aggregated TCCON data as well
($3.29 \pm 0.98$ ppm).

CAMS and CT accurately simulate $CO_2$ growth rate at annual (AGR) and monthly (MGR) resolution for single-site and for global scales. From single-site perspective, the low agreement in MGR estimates between TCCON and both models was found (r < 0.50) only at 3 out of 20 stations. The low agreement was caused by the model-to-observation bias in $CO_2$ concentration during the peak of summer carbon sink season (Tsukuba), by the overwhelming role of ocean carbon fluxes at
the model resolution (Ascension) and to underestimation of $CO_2$ seasonal cycle by the models (Pasadena). These minor MGR caveats have not affected simulated global $CO_2$ growth as CT and CAMS agree with other datasets well at global scales. Namely, perfect TCCON-to-model AGR agreements were observed in NH (r = 0.90 – 0.91), SH (r = 0.96 - 0.97) and tropics (r = 0.95) whereas only reasonable agreement was discerned for global growth (r = 0.65). Then, model-to-GCB agreement is high at global scales (r = 0.74 for CAMS and r = 0.72 for CT). Similarly, model-to-satellite agreement is high
for global growth, in SH and tropics (r = 0.70 – 0.78 for CAMS and r = 0.73 – 0.78 for CT in all these spatial domains) but lower in NH (r = 0.61 for CAMS and r = 0.65 for CT). The modeled AGRs exhibited higher agreement with $AGR_{SAT}$ in SH than in NH which is a minor but interesting finding. This can either indicate (a) a partly erroneous representation of strong $CO_2$ sources (or sinks) from NH by the models that cannot be seen in the model intercomparison but visible while the simulations are compared with observational data or indicate (b) the inability of spaceborne $CO_2$ observations to constrain all
strong $CO_2$ fluxes from NH due to many forfeited measurements in cloudy, hazy and high-latitude regions.

The grid cell correlation between AGR simulations from CAMS and CT (calculated all 3º x 2º cells in the 2004-2016 period) was perfect (median r = 0.99) indicating nearly-ideal agreement in spatial heterogeneity of AGR simulated by the models. It is explainable as the interannual $CO_2$ patterns in the models are on > 90% controlled by their surface fluxes (Schneising et al., 2014). Analysis of land-cover-specific AGR supports this explanation as for most land types correlation in
AGR comparison between CAMS and CT is perfect (r ≥ 0.98). The best agreement of the land-cover-specific AGRs in the models was found for the oceans, snowy surfaces and the regions with no vegetation or with crucial role of ocean carbon fluxes (Antarctica, Oceans and Australia-Oceania). Despite nearly ideal grid-cell correlation between CT and CAMS, the largest discrepancies (0.02 - 0.03 ppm) were found in Middle-West Africa and East Asia. The largest spatial clusters with the most anomalous AGRs (> 20% from global median) were unsurprisingly found by both models in the related regions with
frequent biomass burning (Middle-West Africa) or with intense fossil fuel combustion (East Asia). Since CT and CAMS define a priori biomass burning fluxes using GFED, we surmise that $CO_2$ inverse models with different setups of flux models may exert much higher disagreement in AGR distribution. These discrepancies have not caused any disagreement between





global estimates of AGR by the models since the absolute differences between the CAMS and CT-simulated global AGRs are negligible (< 0.08 ppm, or < 3.9% per year) and temporal agreement is very high (r = 0.97 - 0.99 depending on year).

## 565  5. Conclusions

The results of this study have three vectors of implications. At first, we showed that the modern estimates of annual growth rate of $CO_2$ are consistent across a wide range of datasets. This finding approves the validity of the old approach for calculating $CO_2$ growth rate (Thoning et al., 1989) in terms of no consensus about universal method of $CO_2$ growth rate quantification. At second, the good agreement between multiple datasets in reproducing $CO_2$ atmospheric growth may look
trivial but should not be disregarded in light of the recently reported disagreement between $CO_2$ models that simulate atmospheric $CO_2$ growth (Gaubert et al., 2019). Moreover, the ability of globally-aggregated TCCON data to reproduce global atmospheric $CO_2$ growth auspiciously indicates that many observational gaps of the network are not a critical constraint for quantifying $CO_2$ growth on global scales. Needless to say, that expansion of TCCON network would be still very beneficial (for $CO_2$ growth rate estimation) especially in strong biomass burning or combustion regions where the
models disagreed. The conclusion about TCCON network is important given a few globally-consistent datasets suitable for independent $CO_2$ growth estimation and the weaknesses of the $CO_2$ modeling tools for reproducing independent $CO_2$ growth (Piao et al., 2019). At third, our results have beneficially indicated that intense $CO_2$ release from the regions prone to biomass burning and intense fossil fuel combustion may potentially contribute to the models' disagreement when spatial variability of $CO_2$ growth is analyzed. Supposedly, a sensitivity experiment using $CO_2$ inverse model should be conducted
where alongside with the current setup of a priori biomass burning fluxes, these fluxes could be alternatively defined. Alternative setup could use several independent burned area detection mechanisms that are able to capture small fires more efficiently that it has been done by the current methods (Chuvieco et al., 2019) and could use more realistically-variable burning emissions factors for various types of tropical fires.

**Acknowledgments**

This work was funded by the Korea Meteorological Administration Research and Development Program "Development and Assessment of IPCC AR6 Climate Change Scenario [1365003000]" and was supported by the project [NIMS-2016-3100]. We appreciate the efforts of all researchers who have been studying atmospheric growth of $CO_2$ and interannual variations of carbon. Foremost, we pay tribute to the authors of Buchwitz et al., (2018) whose work was very inspiring for us. We
acknowledge the entire TCCON community for hard work in collecting and processing all these observations around the globe. TCCON data were obtained from the TCCON Data Archive, hosted by CaltechDATA (https://tccondata.org).





CarbonTracker CT2017 results provided by NOAA ESRL, Boulder, Colorado, USA from the website at http://carbontracker.noaa.gov. For providing land cover data we acknowledge Geospatial Information Authority of Japan (Chiba University) and all related collaborating organizations. We also thank the teams responsible for production of CAMS and ONI datasets.

**Author contribution**

Original draft preparation, original idea and data analysis, writing - LDL, manuscript review, editing and methodology development - STK, manuscript review and editing - JK, HL, SL, data support and assisting with editing - YHB, TYG, YSO. All authors have read and agreed to the published version of the current manuscript.

**Competing interests**

The authors declare that they have no conflict of interest.

**Data and code availability**

All the datasets used for this manuscript are available in the open access. All the required links for the data used in this study are provided in the methodology alongside with the date of the last access.

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
