# Peer review of "The consistency between observations (TCCON, surface measurements and satellites) and $CO_2$ models in reproducing global $CO_2$ growth rate"

_Atmospheric Chemistry and Physics, 2020_

## Referee Comment (RC1) · Anonymous Referee #1 · 19 Mar 2020

The manuscript "The consistency between observations (TCCON, surface measurements and satellites) and CO2 models in reproducing global CO2 growth rate" from Labzovskii et al., submitted for publication in Atmos. Chem. Phys., presents and discusses atmospheric CO2 growth rates from different observational data sets and CO2 inverse models. While the topic is in principle important and appropriate for Atmos. Chem. Phys., I see several major shortcomings and cannot recommend publication – at least not without major modifications – as explained in the following.

The authors frequently cite Buchwitz et al., 2018, which is a recent publication addressing essentially the same topic. In the Labzovskii et al. manuscript, the method

of Buchwitz et al., 2018, is used to compute growth rates and also a similar analysis is presented. However, I find the presented analysis quite weak and it remains unclear if and if yes where this publications goes beyond the state-of-the art including the method, results and discussion as presented in Buchwitz et al., 2018.

Also the English needs to be significantly improved. I strongly recommend that the authors consult an expert to improve the English as in the current version there are several errors but often it is also not entirely clear what the authors mean.

In the abstract the authors write: "This study is aimed to advance our knowledge about temporal and spatial variations of annual $CO_2$ growth rate (AGR) by using $CO_2$ observations from the Total Column Observing Network (TCCON), $CO_2$ simulations from Carbon Tracker (CT) and Copernicus Atmospheric Monitoring System (CAMS) models being compared with the previously-reported global references of AGR from Global Carbon Budget (GCB) and satellite observations (SAT) for 2004-2019 years." From the methods used and results presented in the manuscript, I cannot see that the goal to advance our knowledge has been achieved.

In the Conclusions section the authors underline that they have primarily found three results: (i) different $CO_2$ growth rate estimates are consistent, (ii) conclusions w.r.t. modelled and TCCON derived growth rates which I find a bit confusing and (iii) conclusions w.r.t. $CO_2$ from biomass burning and fossil fuel emissions which – I think – are based on a very weak analysis. Concerning (i) I am not aware that inconsistencies had been previously identified as a major issue, which needs to be addressed.

The authors use TCCON data (and this is acknowledged in the Acknowledgements section) but none of the TCCON PIs is a co-author. Have the TCCON PIs been contacted prior to submission of this publication? It would be good to get confirmation that the authors have respected the TCCON data policy (see https://tccondata.org/) and the data policy of the other data sets used in the manuscript.

In the following, I only highlight some aspects, which I think need to be improved. I

could have added more exampled but perhaps this is already sufficient to help the authors to generate a significantly improved manuscript in the future.

Line 113: The authors write: "We present the main tools for retrieving CO2 atmospheric concentration, . . .". This sounds as if the authors have generated atmospheric CO2 data sets but if I understand correctly, they have only used (and analysed) existing data sets. If this is the case than this needs to be clearly stated in Section 2.

Section 2.2: The description of the growth rate computation method is very short and Eq. (1) is unclear (e.g., what is index i / which months are used to compute the growth rate for a given year ?). If the method is (supposed to be) exactly the method of Buchwitz et al., 2018, than this needs to be clearly stated.

Figure 2 and related discussion: I find this figure too busy and therefore a bit confusing. I strongly recommend to limit this figure to panel (a). The ONI / ENSO part should be shown (if really needed for this publication) separately and later in the manuscript. As discussed in Buchwitz et al., 2018, there is a time lag between growth rate changes and ENSO and this important aspect is not appropriately considered here.

A much more detailed presentation and discussion of the TCCON growth rates (shown in, e.g., Fig. 2) needs to be added: Please show detailed results for at least a few representative TCCON sites (XCO2 time series and derived growth rates). How do the growth rates for the different sites compare ? How have the authors dealt with different time periods covered by the different sites ? Much more details on the dependence of the used threshold needs to be added, e.g., it is unclear why 20 is the optimum threshold ? Why not 19 or 21 not shown in (quite sparse) Table 1 ?

Figure 4: The correlation is often quite low, especially for TSU. Is it clear why this is the case ? Is this related to length of time series ?

---

## Short Comment (SC1) · 31 Mar 2020

**TCCON Data Use Policy and License**

Nicholas M. Deutscher[1], Debra Wunch[2], Thorsten Warneke[3], Dietrich G. Feist[4,5], Coleen M. Roehl[6], Voltaire A. Velazco[1], and TCCON Partners[7]

[1]Centre for Atmospheric Chemistry, School of Earth, Atmospheric and Life Sciences, Faculty of Science, Medicine and Health, University of Wollongong, Wollongong, NSW, 2522, Australia
[2]Department of Physics, University of Toronto, Toronto, Ontario, Canada
[3]Institute of Environmental Physics, University of Bremen, Bremen, Germany
[4]Lehrstuhl für Physik der Atmosphäre, Ludwig-Maximilians-Universität München, Munich, Germany
[5]Institut für Physik der Atmosphäre, Deutsches Zentrum für Luft- und Raumfahrt, Oberpfaffenhofen, Germany
[6]Division of Geological and Planetary Sciences, California Institute of Technology, Pasadena, CA, USA

**Correspondence:** Nicholas M. Deutscher (ndeutsch@uow.edu.au)

The paper by Labzovskii et al examining the consistency of growth rates of $CO_2$ amongst different measurement datasets and models makes extensive use of data from the Total Carbon Column Observing Network (TCCON) as one of those measurement datasets. As noted by Anonymous Referee #1 (https://editor.copernicus.org/index.php/acp-2020-114-RC1.pdf?_mdl=msover_md&_jrl=10&_lcm=oc108lcm109w&_acm=get_comm_file&_ms=83680&c=177729&salt=1191195750866577302), the TC-

5  CON Data Use Policy requires that TCCON PIs whose data are used are contacted prior to submission of a manuscript. We note that since the oversight on this specific manuscript, we have been working with the authors to rectify the situation.

The TCCON partners would like to take this opportunity to remind data users of our Data Use Policy and License, and the importance of following similar protocols for other datasets. The important points are:

1. TCCON PIs must be contacted prior to publication or presentation. The contact details of the PIs are in the data file
10    headers. Alternatively, an online form will soon be available, or requests can be emailed to tccon.ftir@gmail.com

2. Co-authorship may be requested, depending on the role of TCCON data and intellectual contribution.

3. TCCON datasets are assigned a digital object identifier (DOI). This must be cited along with any additional appropriate references.

The current Data Use Policy is available at https://tccon-wiki.caltech.edu/Network_Policy/Data_Use_Policy. Importantly,
15  both this and the TCCON Data Archive (www.tccondata.org) refer users to the License in order to understand citation and data use requirements. The License is available as a plain text file with each TCCON data download. An example is included here for the entire TCCON dataset (all sites).

We are currently working to revise the TCCON Data Use Policy and License to make it clearer for data users; however, the general policy will remain unchanged.

20

**TCCON Data License**

1. Downloading and/or using the Data will indicate your acceptance to enter into this license agreement.

3. You have the right to modify and improve on the Data and said modification or improvement will be proprietary to you. In consideration of the royalty-free and collaborative nature of this license agreement, however, you agree to keep the authors informed of the modification and improvement to the Data and hereby grant to TCCON a non-exclusive, unconditional, irrevocable, perpetual and royalty-free right and license to use your modifications or improvements with proper acknowledgement to you as the source.

4. All other rights, including redistribution, display, and publishing adaptations are reserved.

5. At a minimum of 10 weeks prior to publication or presentation of works that include TCCON data, you must contact the individuals listed on the DOI landing page via email and provide a description of the intended publication or presentation. If the TCCON partner makes a significant intellectual contribution to the research, or TCCON data from a site is essential to the work, co-authorship would normally be expected. This should be discussed at an early stage in the work. If the individuals do not respond after 3 emails distributed over the 10 week period, the data are licensed under a cc-by 4.0 license.

6. All Data are assigned a digital object identifier (DOI). Use of these data to support a published work must include a citation that includes the data DOI. Use the standard citation format below, unless a different format is required for publication:

Goo, T.-Y., Y.-S. Oh, V. A. Velazco. 2014. TCCON data from Anmeyondo, South Korea, Release GGG2014R0. TCCON data archive, hosted by CaltechDATA, California Institute of Technology, Pasadena, CA, U.S.A. https://doi.org/10.14291/tccon.ggg2014.anmeyondo01.R0/1149284

Deutscher, N., J. Notholt, J. Messerschmidt, C. Weinzierl, T. Warneke, C. Petri, P. Grupe, K. Katrynski. 2014. TCCON data from Bialystok, Poland, Release GGG2014R1. TCCON data archive, hosted by CaltechDATA, California Institute of Technology, Pasadena, CA, U.S.A. http://doi.org/10.14291/tccon.ggg2014.bialystok01.R1/1183984

Notholt, J., C. Petri, T. Warneke, N. Deutscher, M. Buschmann, C. Weinzierl, R. Macatangay, P. Grupe. 2014. TCCON data from Bremen, Germany, Release GGG2014R0. TCCON data archive, hosted by CaltechDATA, California Institute of Technology, Pasadena, CA, U.S.A. https://doi.org/10.14291/tccon.ggg2014.bremen01.R0/1149275

Griffith, D. W. T., N. Deutscher, V. A. Velazco, P. O. Wennberg, Y. Yavin, G. Keppel Aleks, R. Washenfelder, G. C. Toon, J.-F. Blavier, C. Murphy, N. Jones, G. Kettlewell, B. Connor, R. Macatangay, C. Roehl, M. Ryczek, J. Glowacki, T. Culgan, G. Bryant. 2014. TCCON data from Darwin, Australia, Release GGG2014R0. TCCON data archive, hosted by CaltechDATA, California Institute of Technology, Pasadena, CA, U.S.A. http://doi.org/10.14291/tccon.ggg2014.darwin01.R0/1149290

Iraci, L., J. Podolske, P. Hillyard, C. Roehl, P. O. Wennberg, J.-F. Blavier, J. Landeros, N. Allen, D. Wunch, J. Zavaleta, E. Quigley, G. Osterman, R. Albertson, K. Dunwoody, H. Boyden. 2016. TCCON data from Armstrong Flight Research Center, Edwards, CA, USA, Release GGG2014R1. TCCON data archive, hosted by CaltechDATA, California Institute of Technology, Pasadena, CA, U.S.A. https://doi.org/10.14291/tccon.ggg2014.edwards01.R1/1255068

Strong, K., J. Mendonca, D. Weaver, P. Fogal, J.R. Drummond, R. Batchelor, R. Lindenmaier. 2017. TCCON data from Eureka, Canada, Release GGG2014R2. TCCON data archive, hosted by CaltechDATA, California Institute of Technology, Pasadena, CA, U.S.A. http://doi.org/10.14291/tccon.ggg2014.eureka01.R2

Dubey, M., R. Lindenmaier, B. Henderson, D. Green, N. Allen, C. Roehl, J.-F. Blavier, Z. Butterfield, S. Love, J. Hamelmann, D. Wunch. 2014. TCCON data from Four Corners, NM, USA, Release GGG2014R0. TCCON data archive, hosted by CaltechDATA, California Institute of Technology, Pasadena, CA, U.S.A. https://doi.org/10.14291/tccon.ggg2014.fourcorners01.R0/1149272

Sussmann, R., M. Rettinger. 2014. TCCON data from Garmisch, Germany, Release GGG2014R0. TCCON data archive, hosted by CaltechDATA, California Institute of Technology, Pasadena, CA, U.S.A. https://doi.org/10.14291/tccon.ggg2014.garmisch01.R0/1149299

Iraci, L., J. Podolske, P. Hillyard, C. Roehl, P. O. Wennberg, J.-F. Blavier, J. Landeros, N. Allen, D. Wunch, J. Zavaleta, E. Quigley, G. Osterman, E. Barrow, J. Barney. 2016. TCCON data from Indianapolis, Indiana, USA, Release GGG2014R1. TCCON data archive, hosted by CaltechDATA, California Institute of Technology, Pasadena, CA, U.S.A. https://doi.org/10.14291/tccon.ggg2014.indianapolis01.R1/1330094

Blumenstock, T., F. Hase, M. Schneider, O.E. Garcia, E. Sepulveda. 2014. TCCON data from Izana, Tenerife, Spain, Release GGG2014R1. TCCON data archive, hosted by CaltechDATA, California Institute of Technology, Pasadena, CA, U.S.A. https://doi.org/10.14291/tccon.ggg2014.izana01.R1

Wennberg, P. O., D. Wunch, Y. Yavin, G. C. Toon, J.-F. Blavier, N. Allen, G. Keppel-Aleks. 2014. TCCON data from Jet Propulsion Laboratory, Pasadena, California, USA, Release GGG2014R0. TCCON data archive, hosted by CaltechDATA, California Institute of Technology, Pasadena, CA, U.S.A. http://doi.org/10.14291/tccon.ggg2014.jpl01.R0/1149163

Wennberg, P. O., C. Roehl, J.-F. Blavier, D. Wunch, J. Landeros, N. Allen. 2016. TCCON data from Jet Propulsion Laboratory, Pasadena, California, USA, Release GGG2014R1. TCCON data archive, hosted by CaltechDATA, California Institute of Technology, Pasadena, CA, U.S.A. https://doi.org/10.14291/tccon.ggg2014.jpl02.R1/1330096

Hase, F., T. Blumenstock, S. Dohe, J. Groß, M. Kiel. 2014. TCCON data from Karlsruhe, Germany, Release GGG2014R1. TCCON data archive, hosted by CaltechDATA, California Institute of Technology, Pasadena, CA, U.S.A. https://doi.org/10.14291/tccon.ggg2014.karlsruhe01.R1/1182416

Wennberg, P. O., D. Wunch, C. Roehl, J.-F. Blavier, G. C. Toon, N. Allen, P. Dowell, K. Teske, C. Martin, J. Martin. 2016. TCCON data from Lamont, Oklahoma, USA, Release GGG2014R1. TCCON data archive, hosted by CaltechDATA, California Institute of Technology, Pasadena, CA, U.S.A. https://doi.org/10.14291/tccon.ggg2014.lamont01.R1/1255070

Sherlock, V., B. Connor, J. Robinson, H. Shiona, D. Smale, D. Pollard. 2014. TCCON data from Lauder, New Zealand, 120HR, Release GGG2014R0. TCCON data archive, hosted by CaltechDATA, California Institute of Technology, Pasadena, CA, U.S.A. https://doi.org/10.14291/tccon.ggg2014.lauder01.R0/1149293

Sherlock, V., B. Connor, J. Robinson, H. Shiona, D. Smale, D. Pollard. 2014. TCCON data from Lauder, New Zealand, 125HR, Release GGG2014R0. TCCON data archive, hosted by CaltechDATA, California Institute of Technology, Pasadena, CA, U.S.A. https://doi.org/10.14291/tccon.ggg2014.lauder02.R0/1149298

Dubey, M., B. Henderson, D. Green, Z. Butterfield, G. Keppel-Aleks, N. Allen, J.-F. Blavier, C. Roehl, D. Wunch, R. Lindenmaier. 2014. TCCON data from Manaus, Brazil, Release GGG2014R0. TCCON data archive, hosted by CaltechDATA, California Institute of Technology, Pasadena, CA, U.S.A. https://doi.org/10.14291/tccon.ggg2014.manaus01.R0/1149274

Warneke, T., J. Messerschmidt, J. Notholt, C. Weinzierl, N. Deutscher, C. Petri, P. Grupe, C. Vuillemin, F. Truong, M. Schmidt, M. Ramonet, E. Parmentier. 2014. TCCON data from Orleans, France, Release GGG2014R0. TCCON data archive, hosted by CaltechDATA, California Institute of Technology, Pasadena, CA, U.S.A. https://doi.org/10.14291/tccon.ggg2014.orleans01.R0/1149276

Te, Y., P. Jeseck, C. Janssen. 2014. TCCON data from Paris, France, Release GGG2014R0. TCCON data archive, hosted by CaltechDATA, California Institute of Technology, Pasadena, CA, U.S.A. https://doi.org/10.14291/tccon.ggg2014.paris01.R0/1149279

Wennberg, P. O., D. Wunch, C. Roehl , J.-F. Blavier, G. C. Toon, N. Allen. 2014. TCCON data from California Institute of Technology, Pasadena, California, USA, Release GGG2014R1. TCCON data archive, hosted by CaltechDATA, California Institute of Technology, Pasadena, CA, U.S.A. https://doi.org/10.14291/tccon.ggg2014.pasadena01.R1/1182415

De Maziere, M., M. K. Sha, F. Desmet, C. Hermans, F. Scolas, N. Kumps, J.-M. Metzger, V. Duflot, J.-P. Cammas. 2014. TCCON data from Reunion Island (La Reunion), France, Release GGG2014R0. TCCON data archive, hosted by CaltechDATA, California Institute of Technology, Pasadena, CA, U.S.A. https://doi.org/10.14291/tccon.ggg2014.reunion01.R0/1149288

Morino, I., N. Yokozeki, T. Matzuzaki, A. Shishime. 2017. TCCON data from Rikubetsu, Hokkaido, Japan, Release GGG2014R1. TCCON data archive, hosted by CaltechDATA, California Institute of Technology, Pasadena, CA, U.S.A. https://doi.org/10.14291/tccon.ggg2014.rikubetsu01.R1/1242265

Shiomi, K., Kawakami, S., H. Ohyama, K. Arai, H. Okumura, C. Taura, T. Fukamachi, M. Sakashita. 2014. TCCON data from Saga, Japan, Release GGG2014R0. TCCON data archive, hosted by CaltechDATA, California Institute of Technology, Pasadena, CA, U.S.A. https://doi.org/10.14291/tccon.ggg2014.saga01.R0/1149283

Kivi, R., P. Heikkinen, E. Kyro. 2014. TCCON data from Sodankyla, Finland, Release GGG2014R0. TCCON data archive, hosted by CaltechDATA, California Institute of Technology, Pasadena, CA, U.S.A. https://doi.org/10.14291/tccon.ggg2014.sodankyla01.R0/1149280

Morino, I., T. Matsuzaki, A. Shishime. 2014. TCCON data from Tsukuba, Ibaraki, Japan, 125HR, Release GGG2014R1. TCCON data archive, hosted by CaltechDATA, California Institute of Technology, Pasadena, CA, U.S.A. http://doi.org/10.14291/tccon.ggg2014.tsukuba02.R1/1241486

Griffith, D. W. T., V. A. Velazco, N. Deutscher, C. Murphy, N. Jones, S. Wilson, R. Macatangay, G. Kettlewell, R. R. Buchholz, M. Riggenbach. 2014. TCCON data from Wollongong, Australia, Release GGG2014R0. TCCON data archive, hosted by CaltechDATA, California Institute of Technology, Pasadena, CA, U.S.A. https://doi.org/10.14291/tccon.ggg2014.wollongong01.R0/1149291

Wennberg, P. O., C. Roehl, D. Wunch, G. C. Toon, J.-F. Blavier, R. Washenfelder, G. Keppel-Aleks, N. Allen, J. Ayers. 2014. TCCON data from Park Falls, Wisconsin, USA, Release GGG2014R1. TCCON data archive, hosted by Caltech-DATA, California Institute of Technology, Pasadena, CA, U.S.A. http://doi.org/10.14291/tccon.ggg2014.parkfalls01.R1

Feist, D. G., S. G. Arnold, N. John, M. C. Geibel. 2014. TCCON data from Ascension Island, Saint Helena, Ascension and Tristan da Cunha, Release GGG2014R0. TCCON data archive, hosted by CaltechDATA, California Institute of Technology, Pasadena, CA, U.S.A. http://doi.org/10.14291/tccon.ggg2014.ascension01.R0/1149285

Wunch, D., J. Mendonca, O. Colebatch, N. Allen, J.-F. L. Blavier, S. Springett, D. Worthy, R. Kessler, K. Strong. 2017. TCCON data from East Trout Lake, Canada, Release GGG2014R0. TCCON data archive, hosted by CaltechDATA, California Institute of Technology, Pasadena, CA, U.S.A. https://doi.org/10.14291/tccon.ggg2014.easttroutlake01.R1

Sussmann, R, M. Rettinger. 2018. TCCON data from Zugspitze (DE), Release GGG2014R1. TCCON data archive, hosted by CaltechDATA, California Institute of Technology, Pasadena, CA, U.S.A. https://doi.org/10.14291/tccon.ggg2014.zugspitze01.R1

Notholt, J , T. Warneke, C. Petri, N.M. Deutscher, C. Weinzierl, M. Palm, M. Buschmann. 2017. TCCON data from Ny Ålesund, Spitsbergen (NO), Release GGG2014.R0. TCCON data archive, hosted by CaltechDATA, California Institute of Technology, Pasadena, CA, U.S.A. https://doi.org/10.14291/tccon.ggg2014.nyalesund01.R0/1149278

Morino, I, V.A. Velazco, H. Akihiro, O. Uchino, D.W.T. Griffith. 2018. TCCON data from Burgos, Ilocos Norte (PH), Release GGG2014.R0. TCCON data archive, hosted by CaltechDATA, California Institute of Technology, Pasadena, CA, U.S.A. https://doi.org/10.14291/tccon.ggg2014.burgos01.R0

Liu, C., W. Wang, Y. Sun, 2018. TCCON data from Hefei (PRC), Release GGG2014.R0. TCCON data archive, hosted by CaltechDATA, California Institute of Technology, Pasadena, CA, U.S.A. https://doi.org/10.14291/tccon.ggg2014.hefei01.r0

150 Pollard, D. F., J. Robinson, H. Shiona. 2019. TCCON data from Lauder (NZ), Release GGG2014.R0. TCCON data archive, hosted by CaltechDATA, California Institute of Technology, Pasadena, CA, U.S.A. https://doi.org/10.14291/tccon.ggg2014.lauder03.r0

Failure to comply with the terms of this license will result in forfeit of the right to use this data until the issue is resolved to the satisfaction of the appropriate TCCON PIs.

---

## Short Comment (SC2) · 1 May 2020

**Review and feedback on Labzovskii et al from TCCON PIs**

Nicholas M. Deutscher[1], Debra Wunch[2], Thorsten Warneke[3], Dietrich G. Feist[4,5], Coleen M. Roehl[6], Voltaire A. Velazco[1], Matthias Buschmann[3], Ralf Sussmann[7], Isamu Morino[8], Hirofumi Ohyama[8], Manvendra K. Dubey[9], Rigel Kivi[10], Frank Hase[11], Matthias Schneider[11], Yao Té[12], and David W. T. Griffith[1]

[1]Centre for Atmospheric Chemistry, School of Earth, Atmospheric and Life Sciences, Faculty of Science, Medicine and Health, University of Wollongong, Wollongong, NSW, 2522, Australia
[2]Department of Physics, University of Toronto, Toronto, Ontario, Canada
[3]Institute of Environmental Physics, University of Bremen, Bremen, Germany
[4]Lehrstuhl für Physik der Atmosphäre, Ludwig-Maximilians-Universität München, Munich, Germany
[5]Institut für Physik der Atmosphäre, Deutsches Zentrum für Luft- und Raumfahrt, Oberpfaffenhofen, Germany
[6]Division of Geological and Planetary Sciences, California Institute of Technology, Pasadena, CA, USA
[7]IMK-IFU, Karlsruhe Institute of Technology (KIT), Garmisch-Partenkirchen, Germany
[8]Satellite Observation Center, National Institute for Environmental Studies, Tsukuba, Japan
[9]Earth and Environmental Sciences, Los Alamos National Laboratory, Los Alamos, NM, USA
[10]Space and Earth Observation Centre, Finnish Meteorological Institute, Sodankylä, Finland
[11]IMK-ASF, Karlsruhe Institute of Technology (KIT), Karlsruhe, Germany
[12]Laboratoire d'Etudes du Rayonnement et de la Matière en Astrophysique et Atmosphères (LERMA-IPSL), Sorbonne Université, CNRS, Observatoire de Paris, PSL Université, Paris, France

**Correspondence:** Nicholas M. Deutscher (ndeutsch@uow.edu.au)

The paper by Labzovskii et al examines the consistency of growth rates of $CO_2$ amongst different measurement datasets and models. There are an extensive range of measurements and models included in the analysis; however, we feel that a number of improvements should be made before the paper can be published in ACP. The following comments provide a number of suggestions about how that can be done, with an emphasis on the role and interpretation of TCCON data within the manuscript.

**1 Major Comments**

- The paper does not seem to address the difference between the growth rate in the total column and the growth rate in the surface measurements. There is an important conceptual difference, which could be used to assess vertical mixing in the models.

- In general, it would be beneficial if the statistical results could be supported by some mechanistic analysis and facts about the ground-based locations.

- The method of determining the AGR could be more robust. The simple month(year+1) - month(year) method is prone to uncertainties introduced by short-term variability or (see below) non-temporally uniform sampling. At least using the median in calculating from the monthly growth rates should remove some of the extraneous variability, but the influence of irregular data on the robustness of the calculation of growth rate needs to be investigated. (See point a few below.)

15    – It would be good to include some details about selection criteria for the sites the are used. For example, from a TCCON perspective, why is Ny-Ålesund not included?

– It would be beneficial to include some kind of sampling bias treatment in your analysis to account for the sometimes sparse time series. Some further comments on this follow.

– The paper presents a highly-averaged statistical inter-comparison of TCCON data-inferred atmospheric growth ratio
20    ($AGR_{TCCON}$) and its variability with models and satellite methods. One key finding is that the $AGR_{TCCON}$ correlates with models and satellites only after 2010. This is attributed to the expansion of TCCON after 2009, agreeing well with the references only during the 2010s. This hypothesis should be tested by looking at some specific long-term sites, recognising that effects of the seasonal cycle are important to disentangle from the broader analysis. This can be reduced by picking a southern or low-latitude TCCON, and selecting for a restricted range of solar zenith angles.
25    It would be worth the authors considering the analysis by Lindqvist et al. (2015). In their analysis, they fit TCCON data using a time-varying 6-parameter function for the northern hemisphere. Presenting such a fitting analysis would strengthen the mechanistic interpretation of the paper, for example helping to determine which TCCON sites are best suited to capture variability in the AGRs dominant global effects like ENSO.

– Another important finding is agreement for the large amplitude of the AGR in 2015-2016, during the strongest ENSO.
30    This likely results from the high signal relative to the background in AGR across TCCON. The key point that should be clarified is not only the increase in number of sites, but also the locations of the sites and the timing matters in consistently deriving AGRs. Some fitting of individual TCCON datasets could strengthen this statistical inter-comparison.

– Many TCCON sites are influenced by local/regional effects that would allow differences to be explained, and inform sub-sampling strategies to strengthen your comparisons. For example, at Manaus (Brazil) under the dominant influence
35    of local rainforest, large daily drawdowns (1.8 ppm) occur, which could cause local sink signals on the order of the AGR. Four Corners TCCON site was located near a power plant, which led to early morning plumes that increased xCO2 by up to 10ppm (see and cite Lindenmaier et al. (2014), which could lead to biases in the derived AGR. These effects should be discussed, and e.g. for Four Corners only afternoon data used.
We suggest having a short discussion on the various TCCON site locations, and local/regional effects, which would al-
40    low you to explain differences, define sub-sampling strategies to avoid local/regional effects that bias comparisons, and provide more robust comparisons.
You do discuss some of the model-TCCON site difference in MGR, specifically for Tsukuba, Ascension Island, and Pasadena. However, this discussion and the associated hypotheses are not well backed up by evidence. It also needs to be clear that the failure to represent the MGRs is presumably due to failure to capture something that varies interannually.
45    Regarding the hypotheses, e.g. why would such a small landmass as Ascension Island have an impact at the model res-
olution, especially in total column space? Given its remoteness from large landmass, it would perhaps be expected that Ascension should be among the best represented sites, if we assume that the models get the land source/sinks slightly

wrong. Do the disagreement correlate with particular modules within the models? We do know that Ascension can sometimes be influenced by biomass burning from Africa and/or South America. Perhaps this is a more likely driver of the disagreement. Or it could be that the models don't capture the interannual variability in terrestrial carbon exchange in either Africa or South/Central America - there are few other sites that could be influenced by this.

For Pasadena, the conclusion that carbon uptake at high-latitudes could drive the MGR differences is wholly unconvincing, as there are many sites where the influence of such an effect would be much larger.

At Tsukuba, you don't appear to make any real conclusion, but leave the peak carbon uptake hanging as a possible driver. If that were the case, surely neighbouring sites (Rikubetsu, Saga) would exhibit similar disagreement. The effect might be related more to failing to capture topographical features around the site in the relatively coarse resolution models. There is perhaps a role here for using the satellites to break up the surrounding area to look at potential spatial effects.

- Please include in the introduction as well as late in the TCCON section a brief discussion of the outcome of the work of Yuan et al. (2019), which looks at comparison of in situ, TCCON, satellite measurements and CarbonTracker for $(X)CO_2$

- As indicated in our other comment, please ensure the TCCON data DOIs are appropriately cited for the sites used in this work.

- Uncertainties? There are attempts to interpret sometimes small differences, but no attempt to quantify the uncertainties in the AGR or MGR estimates. In many cases, reporting to 2d.p. (3s.f.) might be excessive.

- Overall, it seems that there are times in the paper where it is a case of the cart driving the horse. Of course it is important and valuable to understand the growth rates and their differences between different datasets, but really what we want to do is use these to diagnose what in our understanding is incomplete and leads to the differences. That's presumably some combination of surface fluxes and atmospheric processes. It is a subtlety, but at times the emphasis throughout the paper of quantifying growth rates is overstated, and should be rephrased to state their importance in interpreting the underlying biogeochemistry/physics.

**2  Minor Comments/Technical Corrections**

The article requires careful proof-reading as there are many grammatical errors. We have attempted to report these, but the list is probably not comprehensive.

- line 9: remove "the" before "global warming"

- line10-11: "Despite atmospheric $CO_2$ growth rate had been considered as the well-known quantity" This wording doesn't work. Maybe "Despite the atmospheric $CO_2$ growth rate having been considered as a well-known quantity"

- line 13: Please correctly define TCCON as the "Total Carbon Column Observing Network"

- line 23-25: The structure of this sentence is confusing, and forms a double negative. It would perhaps be better to emphasise good agreement at 85% of stations.

80
- line 27: "perfect (r=0.99)" - not quite perfect

- line 29: in → a; also again, this is not "perfect".

- line 29: insert "a" before "spatial"

- line 33-36: sentence seems back to front. Missing 'a' before $CO_2$

- line 42: "permanent" - we'd like to hope it isn't permanent, and while perhaps that is optimistic, a better word should be

85
chosen here

- line 43-44, and other locations: when referring to growth rate (GR) it should almost have "the" before it. E.g. here, it should read "The atmospheric $CO_2$ growth rate (GR)". There are many instances of this throughout the manuscript that we will not explicitly point out each time.

- line 46: "the precision of direct observations are high (0.09 ppm)"

90
Please add what kind of instrument has such high precision with citations. In situ measurement? Picarro? Also, "high" is ambiguous, and would perhaps be better replaced by something unambiguous ("excellent") or re-wording ("direct observations are highly precise").

- line 47: "uploaded to" - a better phrasing might be "reported in"

- line 48: at → "on a"

95
- line 50: than → that

- line 51: "constrains" can have a specific meaning in flux estimation, perhaps replace with "limits"

- line 61: insert "the" before terrestrial

- line 67: process → processes

- line 70: insert comma before which, remove "per se"

100
- line 71: ecosystem → ecosystems

- line 72: semiarid → semi-arid

- line 73: "Despite the regions" → "Despite the fact that the regions"

- line 74: evidences → evidence (and are → is)

- line 79: "the limited" → "a limited"

    an alternative is to qualify "the limited number of stations at which measurements are available on long time scales" or something similar

- line 79-81: do we really want to know the growth rate everywhere? You probably want a growth rate for each distinct region (ecosystem or whatever). Your desire to know it everywhere perhaps points to a limitation of the study. Each site's growth rate can be affected by local factors, that may or may not be of interest for the growth rate, and not on a global scale. By selecting a simple 12-month difference method, you leave yourself vulnerable to transient effects from local signals, as does failing to sub-sample measurements to remove local effects. Of course there is a role for both understanding the regional scale effects and changes in local factors as well, but you need to be clear about what you are trying to achieve here.

- line 85: ratio → ratios

- line 88: remove semicolon

- line 96: add "the" before "Global Carbon Budget"

- line 100: "the TCCON" → "TCCON's"

- line 101: "approve" → "prove"

- line 102: "rest" → "remaining"

- line 105: "At second" → "Secondly" (similarly next line for "At third"

- line 108: "Sect.s" → "Sects"

- line 113: Spell out Section here, as you are not explicitly referencing a separate section.

- line 118: the measurements are not continuous because they rely on the presence of sunlight, so by nature cannot measure at night or in cloudy conditions. Quasi-continuous maybe, or ongoing

- line 119: "The XCO2 estimates are retrieved using the ratio"
  XCO2 is not retrieved. Replace with "The XCO2 values are obtained by taking the ratio"

- line 120: $O_2$ (subscript)

- line 121-123: You need references here for these statements.

- line 123: "spectra by pointing on the sun in the near-infrared spectrum" "spectra in the near-infrared region by pointing at the sun"

- line 123: superscript $^{-1}$

- line 125: delete "XCO$_2$"; accuracy → uncertainty (2-sigma)

- line 126: "calibrated" is actually the wrong word here, because it is not truly a calibration. They are compared to these measurements, which links them to the WMO scale

- line 127: "from World Meteorological Organization onboard" "traceable to the World Meteorological Organization scale"

- line 130: exist → existing

- line 131: "familiarize" → "familiarize themselves"

- line 133: "addresses to" → uses

- line 135-137: not clear what this sentence means - needs clarification

- line 145/Figure 1: In this figure, it appears that the green dot "ZUG" should in fact be "KAR" for Karlsruhe. ZUG and GAR are practically co-located and would therefore not be differentiated on this scale map.

- line 150: Insert "The" at the start of the sentence, and "is" before "being"

- line 151: "developed by the Institute of Pierre Simon Laplace (LSCE)"
  LSCE stands for "Laboratoire des Sciences du Climat et de l'Environnement"

- line 157: Sentence starting here needs clarification/rewording.

- line 160: insert "the" before biomass

- line 161: insert "the" before "fire module" and needs clarification on GFAS (GFAS emissions possibly, or "uses the GFAS")

- line 164-165: how did you judge that "this process did not cause serious error propagation"?

- line 175: cover → coverage

- line 177: "CTA"
  What does this stand for? Replace with "CT"?

- line 177-179: please be clear that there are multiple modules for each flux within CarbonTracker. E.g. while CASA is used in both, there are two versions of the biosphere model.

- line 178: "GFED" is already defined in line 161

- line 188, 221, 224: subscript $_2$

- line 191: "being updated at" → "is updated on"

- line 192: why "Obviously"? Suggest removing this word.

- line 195: on → at; insert "The" before GCB

 - line 196: The correct journal name is Earth System Science Data

- line 197-199: Sentence needs to be revised. Either "Large scale AGR estimates... are also taken from" or "Another reference ... is the satellite data" (i.e. delete "are taken from" in this second option)

- line 201: Should be "Greenhouse gases Observing SATellite"

- line 210: is there also a neutral classification (0)?

 - line 226, line 577: Piao et al., 2019 –> Piao et al., 2020

- line 240: about the definition of AGR, MGR was a difference between 2 successive years. But for AGR, isn't it also a difference between 2 successive years? Clarify the difference

- line 252: How do you determine the number of observations per month. Some sites routinely make several hundred per day, so presumably some pre-processing is done.

 - line 252: less → fewer Also, this still seems fairly loose, would it not be better to raise the minimum number of observations contributing to the monthly median? This is discussed later, but there is no reference to that later discussion.

- line 266: "from 1.71 ppm (2009) to 3.35 ppm (2008)" => "from 1.71 ppm (2010 or beginning of 2011 ?) to 3.35 ppm (2009 or beginning of 2010 ?)"

- line 272: do you try selecting the model or satellite data to match the TCCON spatio-temporal measurement pattern?
 This would confirm or refute your assumption.

- line 273 "a study period" → "the study period"

- line 275: r = 0.61 for TCCON-SAT, r = 0.49 for TCCON-GCB In the Abstract and Discussion sections, the correlation co-efficients are described as being 0.75 for TCCON-SAT and 0.68 for TCCON-GCB. Which is correct?

- line 278: "right below" → "directly below the"

 - line 281: not sure why you would expect vegetation-driven seasonality to result in wave-shaped fluctuations. Interannual variability, perhaps, but by the way you are calculating it you are accounting for the seasonal cycles.

- line 283: approve? Not the correct word here

- line 287: "(right axis of Fig. 1, panel b)" → "(right axis of Fig. 2, panel b)"

- line 291: "to data" → "with data"

185    – line 292: remove comma

– line 295: "Besides 15-year pattern" - not sure what you mean?

– line 298: "yield to" → "yield a"

– line 300: This underscores the importance of comparing apples with apples, basically. That is, if you want to make a comparison between similarities in behaviour either across time periods or between datasets, then they need to be
190    sampled to minimize potential spatio-temporal biases.

– Figure 2 and its caption: the scale used for the lines corresponding the AGR_XCO2 is not exactly every 8th month (August) as mentioned in the caption, for example for the peak of the gray line (AGR_TCCON) which is located around 2nd month of 2010.

– Figure 2 caption - missing subscripts on the AGR terms.

195    – line 312: "during the year where periods starting from winter months are shown with green tones, from spring with gray tones, from summer with golden tones, from autumn with red tones." Clarify about differences for different hemispheres. For once, this seems to have a southern hemisphere bias!

– line 319: "be risen" → "arise"

– line 320 "(based on one-time TCCON observations)"
200    Is it one measurement of TCCON or is it a daily mean measurement of TCCON?

– line 321-322: Not sure what this sentence means - clarify.

– line 329: "different thresholds we mentioned above" → "the different thresholds mentioned above"

– line 332-335: perhaps it would be better to correlate the number of available MGR with the ratio of TCCON to SAT (or model) error spreads, or again compare by subsampling the model or satellite data to resemble the relevant TCCON data
205    for each period.

– line 347: such as 30 points

– line 349: approves → proves (?)

– line 363: versus → with

– line 367: add "the" before "correlation"

210    – line 378: plain → simplistic (assuming this is what you mean)

- line 381: regarding the Tsukuba site - probably the topography cannot be resolved at these model resolutions. Also, if the difference between models and TCCON is driven by the land carbon sink as you hypothesise, surely the neighbouring sites (Rikubetsu, Saga) should exhibit similar behaviour.

- line 387: It's not "most likely" that Ascension is a small island! Please fix this wording. Though as noted in the general comments it is not clear that this small landmass would have affect model-measurement differences here in the regionally-representative column.

- line 397: add "themself" after familiarize

- line 400/Figure 4: It looks like CAMS is missing for WOL, and CT missing for ASC. It would be helpful to include the values for the co-efficients on the plot.

- line 415: on → by?
  Seems fundamentally like the average AGRs should be in good agreement, unless there is something wrong with a dataset, or the comparative data are sampled so as to introduce biases between them. This is again an instance where the models/satellites should be sampled in the same spatio-temporal fashion as TCCON before drawing any conclusions about comparative AGRs.

- line 416: From → "From a"

- line 420: this is not perfect correlation.

- Figure 5: In the "Global" panel (top right side), there are TCCON data for 2006 and 2007 (gray bars), but there aren't any in the 3 other panels (neither in "NH", nor in "SH" or "Tropical"). Maybe emphasise that there must be a minimum 2(?) sites contributing within each region.

- line 434: variability → coverage or representation

- line 436: dimensions? Do you mean on different scales?

- line 438: errors → differences

- line 439: it's not clear how the time lag affects model AGR differences, unless there are differences in the models capturing this.

- line 446: citations needed here

- line 454: again, not "perfect" agreement

- line 456: Oceans doesn't need to be capitalised

- line 474-475: Not sure what this sentence means.

- line 476: that → as

240     – line 477-481: lots of reporting of statistics without any need. This could be simplified.

- line 492: approves → supports

- line 493: not perfect

- line 498: "oppositely" - suggest replacing with an alternative word.

- line 501: at → in the

245     – line 504: "of by" → "by"

- line 511: again, not perfect

- line 515/Table 2: Please add units for "Difference".

- Table 2 and lines 493-500: The median differences and correlation co-efficients are incosistent between the table and the text.

250     – line 530: agreements → agreement

- line 538-540: revise as appropriate once earlier section is revisited

- line 542: another misuse of "perfect", and in this case I wouldn't even call them near-perfect.

- line 552: not perfect

- line 553: sentence needs revising ("on > 90%" doesn't make sense)

255     – line 555: not perfect

- line 558: 0.02-0.03ppm → 0.04 ppm? (c.f. line 494 and 495)

- line 566: "The results of this study have three vectors of implications." What does this mean? Why not just say there results have three implications?

- Conclusions: as noted earlier, "at second" and "at third" should be replaced by "secondly" and "thirdly"

260     – line 567: approves → confirms; old → existing

- supplement, line 8: Table S.2.2 → Table S.2.1

- supplement, line 64, 85: subscript $_2$

**References**

Lindenmaier, R., Dubey, M. K., Henderson, B. G., Butterfield, Z. T., Herman, J. R., Rahn, T., and Lee, S.-H.: Multiscale observations of
CO2, 13CO2, and pollutants at Four Corners for emission verification and attribution, Proceedings of the National Academy of Sciences,
111, 8386–8391, https://doi.org/10.1073/pnas.1321883111, http://www.pnas.org/content/111/23/8386.abstract, 2014.

Lindqvist, H., O'Dell, C. W., Basu, S., Boesch, H., Chevallier, F., Deutscher, N., Feng, L., Fisher, B., Hase, F., Inoue, M., Kivi, R., Morino, I.,
Palmer, P. I., Parker, R., Schneider, M., Sussmann, R., and Yoshida, Y.: Does GOSAT capture the true seasonal cycle of carbon dioxide?,
Atmospheric Chemistry and Physics, 15, 13 023–13 040, https://doi.org/10.5194/acp-15-13023-2015, https://www.atmos-chem-phys.net/
15/13023/2015/, 2015.

Yuan, Y., Sussmann, R., Rettinger, M., Ries, L., Petermeier, H., and Menzel, A.: Comparison of Continuous In-Situ CO2 Measurements with
Co-Located Column-Averaged XCO2 TCCON/Satellite Observations and CarbonTracker Model Over the Zugspitze Region, Remote
Sensing, 11, 2981, https://doi.org/10.3390/rs11242981, http://dx.doi.org/10.3390/rs11242981, 2019.

---

## Referee Comment (RC2) · Anonymous Referee #2 · 15 Jun 2020

General comments:

In their work "The consistency between observations (TCCON, surface measurements and CO2 models in reproducing global CO2 growth rate", submitted to ACP, the authors investigate the agreement of the atmospheric CO2 (annual) growth rates from several data sources: total column CO2 from the network of ground-based Fourier Transform Spectrometers (TCCON), inverse model estimates from Carbon-Tracker 2017 and CAMS, and reported growth rates from the Global Carbon Budget as well as total column CO2 from satellites. While quantifying the atmospheric CO2 growth rate using different data sources might be a topic suitable for ACP, the scientific

questions that this paper attempts to address are not sufficiently clearly defined. Thus, the research presented in the paper seems to somewhat suffer from this throughout the manuscript. The concepts, ideas, tools and data in the paper are not novel, and the methods used for data analysis are not entirely valid or sufficiently described. I consider that the paper should not be accepted for publication in its current form. A thorough revision starting from re-formulating the research questions would be necessary. In what follows, I will point out more specifically my concerns and comments on the manuscript, focusing more on the early sections as they naturally affect the rest of the manuscript. Because the comments are quite extensive, I will exceptionally not list technical corrections nor specific comments related to the English language in my review.

Specific comments:

Lines 64-89: The main message of this paragraph is unclear to the reader: are we interested in the global or local CO2 growth rate in this paper?

The authors start by highlighting the need for an accurate knowledge of the global CO2 growth rate; however, their analysis focuses on both local and global analysis without a clear focus or guidance to what essentially is important in the results and why the analysis has been made. On lines 97-104, the authors state the three objectives of their study. Regarding these objectives (a)-(c), (a) I don't think that aggregating all TCCON data would represent the global CO2 growth rate because of several reasons: even though the TCCON is in principle a (near-)global network, the instruments are mainly located in North America and Europe. In addition, several of them are affected by local sources (e.g., Paris, Tsukuba, Pasadena) that make them interesting for the evaluation of satellite CO2 retrievals in urban circumstances but maybe less representative of the global CO2 background for the purposes of this study. Regarding (b) and (c), the estimation of spatiotemporal inconsistencies between inverse models has already been carried out in a number of studies. It is not clear whether this study brings anything new to the discussion. In particular, I found the analysis regarding (c) a bit rushed and

shallow, and lacking important references to earlier studies that consider either these particular models or regions of interest (e.g., Lindqvist et al., 2015; Palmer et al., 2019).

The analysis focuses on the atmospheric CO2 growth rate and several times mentions a "permanent" growth of CO2, which may be misleading to readers, considering the seasonal cycle of CO2.

Section 2.1.1 (TCCON): The TCCON data policy requires that the authors contact the TCCON PI's in the preparation phase of the paper in order to guarantee that the data are used and interpreted correctly and also to agree on a potential co-authorship in case the TCCON data have a central role in the manuscript. Since the TCCON PI's have already commented on this issue separately, I do not focus on this more. I do want to add, however, that several issues in the manuscript regarding the interpretation of the results at specific TCCON sites would have been clarified in the preparation process of the manuscript in case the TCCON PIs had been contacted for the work.

Sections 2.1.2, 2.1.3, and 2.1.4: These sections lack plenty of relevant details, such as version numbers of several of the prior flux components and a more detailed description of the satellite data, even though these data were cited ("CO2 observations from SCIAMACHY and GOSAT" is not sufficient).

Section 2.2.1: Description of the methodology is not sufficient for reproducing the results. For example, it is not clearly described how the gaps in the data are considered. Are there any criteria for including or excluding some of the TCCON sites? Exact methodologies should also be described for comparisons of model and TCCON data (e.g., spatiotemporal interpolation of the gridded model data, averaging kernel corrections etc.).

The results and discussion sections suffer from a very scattered analysis which is rich in details but not in content, and lacks focus. Correlation analysis is not sufficient in case of time series: for example, a phase difference in the time series would result in a relatively weak correlation but the reason for the weak correlation would not be

identified. At least some representative cases of the XCO2 time series should be presented. The discussions and conclusions drawn on the claimed "biomass burning regions" seem particularly rushed and would have been relatively straightforward to check by the authors because at least CarbonTracker 2017 provides the imposed fire fluxes as a separate data field.

---

## Author Comment (AC2) · 5 Dec 2020

**RESPONSE TO THE REVIEWER #1**

The manuscript "The consistency between observations (TCCON, surface measurements and satellites) and $CO_2$ models in reproducing global CO2 growth rate" from Labzovskii et al., submitted for publication in Atmos. Chem. Phys., presents and discusses atmospheric $CO_2$ growth rates from different observational data sets and $CO_2$ inverse models. While the topic is in principle important and appropriate for Atmos. Chem. Phys., I see several major shortcomings and cannot recommend publication – at least not without major modifications – as explained in the following.

Response 1.01: We thank the reviewer for the valuable suggestions based on which we have substantially improved the manuscript. Point-by-point responses are given below and marked by green color. Please note that the definition of GR, AGR and MGR terms are given in the article. We underline that our manuscript has undergone major modifications based on versatile comments/suggestions of the reviewers and also by the TCCON community (see their collective comment at the ACP page of this revision).

The authors frequently cite Buchwitz et al., 2018, which is a recent publication addressing essentially the same topic. In the Labzovskii et al. manuscript, the method of Buchwitz et al., 2018, is used to compute growth rates and also a similar analysis is presented. However, I find the presented analysis quite weak and it remains unclear if and if yes where this publications goes beyond the state-of-the art including the method, results and discussion as presented in Buchwitz et al., 2018.

Response 1.02: Due to prevalence of TCCON-oriented analysis and according to the comments from the reviewers, we reformulated the research aim and the objectives of the manuscript.

- New research aim is, to quote *"This study aims to understand whether TCCON aggregated observations are currently suitable for robust estimation of the global $CO_2$ growth rate given the existing spatio-temporal gaps of the network"*
- Updated objectives are *"(a) to estimate the robustness of $AGR_{TCCON}$ due to the data sampling, measurement gaps or difference in time series across the sites"*. Secondly, (b) *"to examine the $AGR_{TCCON}$ agreement with the existing $CO_2$ growth references and its sensitivity to external factors"*. Additional objective includes (c) *"to assess the exposure of $CO_2$ growth estimates at each TCCON station to external factors"*

This publication does not present any new method since we use Buchwitz et al., (2018) as the methodological reference for AGR calculation. We made it more clear in the manuscript by stating *"More specifically, we use exactly the methodology of Buchwitz et al. (2018) whereas their method originates from the commonly-accepted GR calculation approach (Thoning et al., 1989)"* in section 2.2.1. Meanwhile the main new finding is that *"We showed that the current estimates of $CO_2$ annual growth obtained*

*from the TCCON aggregated observations are adequate and are in reasonable agreement with the existing references even when the simple methodology (Thoning et al., 1989) applied."* This main finding is mentioned in the conclusion where the new implications from the study findings are given in the second part of conclusions (after line 956).

Also the English needs to be significantly improved. I strongly recommend that the authors consult an expert to improve the English as in the current version there are several errors but often it is also not entirely clear what the authors mean. In the abstract the authors write: "This study is aimed to advance our knowledge about temporal and spatial variations of annual CO2 growth rate (AGR) by using CO2 observations from the Total Column Observing Network (TCCON), CO2 simulations from Carbon Tracker (CT) and Copernicus Atmospheric Monitoring System (CAMS) models being compared with the previously-reported global references of AGR from Global Carbon Budget (GCB) and satellite observations (SAT) for 2004-2019 years."

Response 1.03: We applied the suggestions for language corrections from the TCCON community collective feedback. As several TCCON PIs signed this document are native English speakers, we hope that the English language is significantly improved. The new experienced coauthors included in this study have hopefully improved the level of English language as well. If the level of English remains unsatisfactory at this stage, at the next stage of revision, we will either ask one of our native English-speaking coauthors or editing company for thorough grammar and stylistic check of the article.

From the methods used and results presented in the manuscript, I cannot see that the goal to advance our knowledge has been achieved.

Response 1.04: We agree that the previous study goal was too vaguely formulated (see Response 1.02 for new research aim and objectives).

In the Conclusions section the authors underline that they have primarily found three results: (i) different CO2 growth rate estimates are consistent, (ii) conclusions w.r.t. modelled and TCCON derived growth rates which I find a bit confusing and (iii) conclusions w.r.t. CO2 from biomass burning and fossil fuel emissions which – I think – are based on a very weak analysis.

Response 1.05: The conclusions are now based on three reformulated objectives.

- *(a) "to estimate the robustness of $AGR_{TCCON}$ due to the data sampling, measurement gaps or difference in time series across the sites".* This objective is now supported by expanded evidences about daily, monthly, annual and seasonal stability of TCCON AGR estimates, objective
- *(b) "to examine the $AGR_{TCCON}$ agreement with the existing $CO_2$ growth references and its sensitivity to external factors".*

- This is basically similar to what was stated for objective (i) from the reviewer's comment about but complemented with more results (Figure 7, Figure 9, Figure 10, Figure 11, Figure S.2.4).
- We agree that (iii) objective and conclusions were based on weak analysis. We reformulated this objective to be (c) *"to assess the exposure of $CO_2$ growth estimates at each TCCON station to external factors"*. We included an extended analysis of fluxes from CarbonTracker (Figure 14) and their relationship with the model disagreement. Moreover, we analyzed whether the model disagreement was driven by the vertical mixing (Figure 15 where model-to-surface $CO_2$ data is analyzed).

Concerning (i) I am not aware that inconsistencies had been previously identified as a major issue, which needs to be addressed.

Response 1.06: We agree that this point should not be the main factor driving the research from our study. We reformulated the main motivation of this study as *"The $CO_2$ growth rate (GR) is one of the key geophysical quantities reflecting the dynamics of the climate change but there are still few global observational approaches for quantifying global GR"* which is given in the abstract. However, some inconsistencies between different models in reproducing $CO_2$ growth rate do exist according to Gaubert et al., (2019) reference given in this study.

The authors use TCCON data (and this is acknowledged in the Acknowledgements section) but none of the TCCON PIs is a co-author. Have the TCCON PIs been contacted prior to submission of this publication? It would be good to get confirmation that the authors have respected the TCCON data policy (see https://tccondata.org/) and the data policy of the other data sets used in the manuscript.

Response 1.07: We did have TCCON PI even in the first version of the manuscript (Young-Suk Oh plus Taeyoung Goo from the Anmyeondo measurement site). Most importantly, the oversight with the TCCON policy compliance has been alleviated as during the revision we had been contacted by the TCCON co-chair, Asia/Oceania (Nicholas Deutscher). Under his assistance, we followed the TCCON Data License and policies and shared the material of the manuscript with all PIs of the TCCON network and those who were interested, provided their interactive comments. We have also added the missing references required by the TCCON policies to the current version. Moreover, we have offered coauthorship to the TCCON researchers responsible for composing TCCON collective feedback. However, so far all the TCCON researchers decided to keep providing comments without becoming coauthor for this manuscript.

In the following, I only highlight some aspects, which I think need to be improved. I could have added more exampled but perhaps this is already sufficient to help the authors to generate a significantly improved manuscript in the future.

Line 113: The authors write: "We present the main tools for retrieving CO2 atmospheric concentration, . . .". This sounds as if the authors have generated atmospheric CO2 data sets but if I understand correctly, they have only used (and analysed) existing data sets. If this is the case than this needs to be clearly stated in Section 2.

Response 1.08. This formulation is removed. In Section 2 we state, before describing all datasets "*It should be noted that this study does not generate any new $CO_2$ data and relies on the referenced $CO_2$ observations or simulations from the existing sources.*"

Section 2.2: The description of the growth rate computation method is very short and Eq. (1) is unclear (e.g., what is index i / which months are used to compute the growth rate for a given year ?).

Response 1.09: Please see new, more detailed formulation based on Equations 3 and 4 from the Section 2.2.1. Regarding which months should be used, see the quote from the study *"In an ideal position, there would be 12 MGRs as an input for each TCCON station (so MGR is available for every month from January to December)"*.

If the method is (supposed to be) exactly the method of Buchwitz et al., 2018, than this needs to be clearly stated.

Response 1.10: We use exactly Buchwitz et al., (2018) calculation and we once again emphasized it by stating *"In this regard, we refer to the latest outcomes from the GR research, and we use exactly the methodology of Buchwitz et al. (2018)"* in Section 2.2.1.

Figure 2 and related discussion: I find this figure too busy and therefore a bit confusing. I strongly recommend to limit this figure to panel (a).

Response 1.11: Done

The ONI / ENSO part should be shown (if really needed for this publication) separately and later in the manuscript. As discussed in Buchwitz et al., 2018, there is a time lag between growth rate changes and ENSO and this important aspect is not appropriately considered here.

Response: 1.12 ONI/ENSO part is moved to other parts of the manuscript where we investigated the agreement between AGR and ENSO strength (Figure 9). We have also considered $\pm$ 1-year time lag for this analysis (also Figure 9). Moreover, we investigated the role of ONI on the disagreement rate between $AGR_{TCCON}$ and the references (Figure 10, Figure 11).

A much more detailed presentation and discussion of the TCCON growth rates (shown in, e.g., Fig. 2) needs to be added: Please show detailed results for at least a few representative TCCON sites (XCO2 time series and derived growth rates).

Response 1.13: Please see the analysis of the few representative TCCON sites (Tsukuba, Park Falls, Garmisch) included in the station-wise analysis (Section 3.2.1). Additional analysis is given for Ascension measurement site in the supplementary material as well.

How do the growth rates for the different sites compare?

Response 1.14: We included a new section (3.2.2 with Figure 5) dedicated to this intercomparison.

How have the authors dealt with different time periods covered by the different sites ?

Response 1.15: We analyzed the role of daily, sub-monthly and annual variability of $XCO_2$ and their role on single-station $AGR_{TCCON}$ estimates. See the second part of section 3.2.2 and the description provided for Table 1 where the data abundance statistics for each TCCON site are reported.

Much more details on the dependence of the used threshold needs to be added, e.g., it is unclear why 20 is the optimum threshold ? Why not 19 or 21 not shown in (quite sparse) Table 1 ?

Response 1.16: Please see extended analysis from Section 3.1 (Figure 2) where we analyzed all possible daily thresholds could be applied for $XCO_2$ daily data from TCCON station. The error spread due to the daily threshold is now reported on two figures (Figure 3, Figure 7). We agree that there is no optimum threshold for the current methodology. We applied '2' as the daily threshold because the agreement with the references is the same like for stringent thresholds but we have more data to analyze. To confirm, *"First of all, the change of daily threshold from '2' to higher values (3-25) does not lead to improved correlation of $AGR_{TCCON}$ with either SAT (r ⩽ 0.60 for any threshold) or GCB (r ⩽ 0.56 for any threshold). At one hand, these findings confirm the correctness of our approach of saving as much data as possible by using daily threshold '2'. At other hand, it recognizes once again that the disagreement is not driven by data quality or sub-monthly structure."*

Figure 4: The correlation is often quite low, especially for TSU. Is it clear why this is the case ? Is this related to length of time series ?

Response 1.17: It is unlikely due to the time series length (see Figure 4 and the respective section 3.2.1). We found interesting pattern that is related to the vicinity to

large cities (see section 3.4.2, Table 2 and also Figures S.2.4 and S.2.5 in the supplementary for the additional information).

---

## Author Comment (AC3) · 5 Dec 2020

**RESPONSE TO THE TCCON COMMENTS**

We thank the TCCON community and the PIs for the valuable suggestions. Point-by-point responses are given below and marked by green color. For brevity, we respond "OK" to the comments that were precisely applied according to the recommendation and "reformulated" to the comments where we had to rethink about new formulation according to the less specific comment. Note that the definition of GR, AGR and MGR terms are given in the article.

– The paper does not seem to address the difference between the growth rate in the total column and the growth rate in the surface measurements. There is an important conceptual difference, which could be used to assess vertical mixing in the models.

Response 3.01: We have incorporated this analysis (see Figure 12 and the corresponding section).

– In general, it would be beneficial if the statistical results could be supported by some mechanistic analysis and facts about the ground-based locations. $\sigma_a$

Response 3.02: We have added the description of TCCON sites used in this study (second paragraph of Section 2.1.1). Analysis of selected TCCON sites is added (section 3.2.1).

– The method of determining the AGR could be more robust. The simple month(year+1) - month(year) method is prone to uncertainties introduced by short-term variability or (see below) non-temporally uniform sampling. At least using the median in calculating from the monthly growth rates should remove some of the extraneous variability, but the influence of irregular data on the robustness of the calculation of growth rate needs to be investigated. (See point a few below.)

Response 3.03:

- Since our paper is not aimed on the intercomparison of different methods for AGR, we strictly followed Buchwitz et al., (2018) procedure as the robust method. However, we have also acknowledged the raised issue from your comment as *"AGR may considerably vary depending on the methodology of calculation (Piao et al., 2020)"*.
- We did use median in calculating the monthly growth rates for every year. Basically, what the TCCON reviewer called "The simple month(year+1) - month(year) method" is MGR shown by Equation 3. Meanwhile, *"median in calculating from the monthly growth rate"* is basically AGR shown in Equation 4.
- The influence of irregular data is thoroughly investigated in this version of the article (see objective (a) of this study and the analysis for Figure 2, upper panel of Figure 3, Table 1, orange area of Figure 7)

– It would be good to include some details about selection criteria for the sites that are used. For example, from a TCCON perspective, why is Ny-Ålesund not included?

Response 3.04: Ny-Alesund is not completely excluded from the article (see Figure 5 and Figure 12 for instance). It is missing on Figure 6 due to specific reason *"ERK and NYA stations are not used in this analysis due to lack of data for evaluating seasonal cycle according to LM."* where LM is Lindqvist et al., (2015) methodology. All details for filtering out TCCON stations are given on Figure 2 and section 3.1.

– It would be beneficial to include some kind of sampling bias treatment in your analysis to account for the sometimes-sparse time series. Some further comments on this follow.

– The paper presents a highly-averaged statistical inter-comparison of TCCON data-inferred atmospheric growth ratio (AGRTCCON) and its variability with models and satellite methods. One key finding is that the AGRTCCON correlates with models and satellites only after 2010. This is attributed to the expansion of TCCON after 2009, agreeing well with the references only during the 2010s. This hypothesis should be tested by looking at some specific long-term sites, recognizing that effects of the seasonal cycle are important to disentangle from the broader analysis. This can be reduced by picking a southern or low-latitude TCCON, and selecting for a restricted range of solar zenith angles. It would be worth the authors considering the analysis by Lindqvist et al. (2015). In their analysis, they fit TCCON data using a time-varying 6-parameter function for the northern hemisphere. Presenting such a fitting analysis would strengthen the mechanistic interpretation of the paper, for example helping to determine which TCCON sites are best suited to capture variability in the AGRs dominant global effects like ENSO.

Response 3.05.

- We analyzed the extended time series including 2018 and the hypothesis about the period-driven agreement is rejected in this version. See the new explanation of the TCCON-to-reference agreement part (Section 3.3).
- Please note that we have invited Hannakaisa Lindqvist to our manuscript. She helped us to make a comparison between the AGR and the linear trend of $XCO_2$ using their methodology (Lindqvist et al., 2015 paper). See Figure 6 please. It did help providing mechanistic explanation of which sites are more independent from the effects of time series edge/seasonal maximum $CO_2$ effect.
- The key sampling bias is the lack of station-wise AGR in the global AGR estimates. This irregularity is reflected at the error bars plotted using data abundance-driven uncertainties using *"average standard deviation across TCCON stations (AGR) multiplied on $\sqrt{N_{total}}/N$ factor where N – number of*

*stations used, $N_{total}$ – total number of stations in TCCON analysis."* This calculation goes in line with Buchwitz et al., (2018) methodology.

– Another important finding is agreement for the large amplitude of the AGR in 2015-2016, during the strongest ENSO. This likely results from the high signal relative to the background in AGR across TCCON. The key point that should be clarified is not only the increase in number of sites, but also the locations of the sites and the timing matters in consistently deriving AGRs. Some fitting of individual TCCON datasets could strengthen this statistical inter-comparison.

Response 3.06: Please see new Figure 6 for the fitting analysis of individual TCCON sites. We included the phrase *"In future, not only the increase in number of sites but also the locations of the TCCON sites and the timing of observations would be crucial for deriving robust $CO_2$ growth rates."* in the conclusions.

– Many TCCON sites are influenced by local/regional effects that would allow differences to be explained, and inform sub-sampling strategies to strengthen your comparisons. For example, at Manaus (Brazil) under the dominant influence of local rainforest, large daily drawdowns (1.8 ppm) occur, which could cause local sink signals on the order of the AGR.

Response 3.07: Manaus station is not included in the analysis due to lack of MGR for the period of study (see Figure 2 and the corresponding section).

Four Corners TCCON site was located near a power plant, which led to early morning plumes that increased xCO2 by up to 10 ppm (see and cite Lindenmaier et al. (2014), which could lead to biases in the derived AGR. These effects should be discussed, and e.g. for Four Corners only afternoon data used.

Response 3.08:
- Four Corners has too short time series to be included in the analysis (see the response 3.07 above for the details).
- We have included the investigation of the role of neighboring urban sources in the manuscript using MODIS urban pixels (see section 3.4.2 please). However, this method does not allow tackling strong non-urban sources such as power plants since standalone power plant facilities are likely missing in the MODIS urban data. Due to this, we added we included the warning about this in the conclusions *"strong $CO_2$ sources are not always attributed to built-up zones with clear spatial extent and structure such as megacities. They can be attributed to emissions from a power plant as it was shown in the previous studies (Lindenmaier et al., 2014)".*

We suggest having a short discussion on the various TCCON site locations, and local/regional effects, which would allow you to explain differences, define sub-sampling strategies to avoid local/regional effects that bias comparisons, and provide more robust comparisons.

Response 3.09:

- We have included discussion of $XCO_2$, MGR and AGR series for several selected sites in section 3.2.1 (Figure 4).
- Since the role of data abundance/sampling/irregularity for AGR calculation is pivotal, we have added a separate objective to investigate this issue. Namely, (a) *"to estimate the robustness of $AGR_{TCCON}$ due to the data sampling, measurement gaps or difference in time series across the sites".*
- The (a) objective is now supported by extended evidences about daily, monthly, annual and seasonal stability of TCCON AGR estimates (see 2nd paragraph of the discussion for the details). Since we found low influence of daily, monthly and annual data abundance (and seasonality) on MGR and AGR estimates, we did not apply more sophisticated data sampling strategy and concluded that *"current estimates of $CO_2$ annual growth obtained from the TCCON aggregated observations is adequate and are in reasonable agreement with the existing references even when the simple methodology (Thoning et al., 1989) and the simple TCCON data screening are applied"* (see conclusions).

You do discuss some of the model-TCCON site difference in MGR, specifically for Tsukuba, Ascension Island, and Pasadena. However, this discussion and the associated hypotheses are not well backed up by evidence. It also needs to be clear that the failure to represent the MGRs is presumably due to failure to capture something that varies interannually. Regarding the hypotheses, e.g. why would such a small landmass as Ascension Island have an impact at the model resolution, especially in total column space? Given its remoteness from large landmass, it would perhaps be expected that Ascension should be among the best represented sites, if we assume that the models get the land source/sinks slightly wrong. Do the disagreement correlate with particular modules within the models? We do know that Ascension can sometimes be influenced by biomass burning from Africa and/or South America. Perhaps this is a more likely driver of the disagreement. Or it could be that the models don't capture the interannual variability in terrestrial carbon exchange in either Africa or South/Central America - there are few other sites that could be influenced by this. For Pasadena, the conclusion that carbon uptake at high-latitudes could drive the MGR differences is wholly unconvincing, as there are many sites where the influence of such an effect would be much larger.

Response 3.10:

- Note that Pasadena and Tsukuba early hypotheses are now rejected based on the new evidences.
- We also think that MGR can exhibit disagreement due to exposure of TCCON site to some localized influence which cannot be captured by models.
- This hypothesis was checked by analyzing the urban influence on interannual TCCON signal is expressed by the agreement rate of MGR estimates between TCCON and the models (section 3.4.2). TCCON-to-model correlation coefficient has negative agreement (r = - 0.73) with the size of the closest megacity to TCCON station (calculated by MODIS urban pixels, megacity = city > 1500 km$^2$). Since it can indicate a potential exposure of TCCON station to urban $CO_2$ emissions, we approximately quantified this exposure by using the distance to the closest megacity. According to this analysis, Paris, Tsukuba, Saga, Pasadena and Karlsruhe are theoretically the most influenced stations (< 40 km to megacity). However, the reviewer concern about malignant role of these sites in global AGR signal is not major factor here since "The difference between original AGR$_{TCCON}$ and AGR$_{TCCON}$ without "the most urbanized sites" ranged from negligibly low ~0.00 ppm to 0.29 ppm (2017) despite these "most urbanized sites" composed >20% of observational cover of TCCON in 2017."
- Thank you for valuable comment about Ascension as this hypothesis about biomass burning is likely the reason of disagreement. Namely, we included the analysis of biomass burning fluxes using CT (see Figure 14 and the associated description in the paragraph above). Also, detailed time series of XCO$_2$, MGR and AGR from Ascension is added to supplement (Figure S.2.7).

At Tsukuba, you don't appear to make any real conclusion, but leave the peak carbon uptake hanging as a possible driver. If that were the case, surely neighboring sites (Rikubetsu, Saga) would exhibit similar disagreement. The effect might be related more to failing to capture topographical features around the site in the relatively coarse resolution models. There is perhaps a role here for using the satellites to break up the surrounding area to look at potential spatial effects.

Response 3.11: As mentioned in the response 3.10, we have not found evidences for the role of Tsukuba, Saga and Rikubetsu. The explanation of this disagreement is given in the same response 3.10. There is no high correlation between TCCON-to-model correlation coefficient (for MGR) and the altitude of the site. Despite this and the finding about urban influence, we incorporated the TCCON reviewer's suggestion as *"It should be noted that except the urban influence, the disagreement at some TCCON sites can be related to failing to capture topographical features around the site in the relatively coarse resolution models."*

– Please include in the introduction as well as late in the TCCON section a brief discussion of the outcome of the work of Yuan et al. (2019), which looks at comparison of in situ, TCCON, satellite measurements and CarbonTracker for (X)CO2

Response 3.12: Yuan et al. (2019) description is included in the introduction (3$^{rd}$ paragraph) with the following reference:

Yuan, Y., Sussmann, R., Rettinger, M., Ries, L., Petermeier, H., and Menzel, A.Comparison of Continuous In-Situ $CO_2$ Measurements with Co-Located Column-Averaged $XCO_2$ TCCON/Satellite Observations and CarbonTracker Model Over the Zugspitze Region, Remote Sens. **2019**, 11, 2981; doi:10.3390/rs11242981

– As indicated in our other comment, please ensure the TCCON data DOIs are appropriately cited for the sites used in this work.

Response 3.13: TCCON data DOIs are cited for all sites used in this work.

– Uncertainties? There are attempts to interpret sometimes small differences, but no attempt to quantify the uncertainties in the AGR or MGR estimates. In many cases, reporting to 2d.p. (3s.f.) might be excessive.

Response 3.14:

- We estimated the error spread of MGR input into the AGR (Figures 3 and 7). As only the threshold of "2" measurements is used in the manuscript, this estimate cannot be included in the error propagation but serves a good illustration for potential error spread could be caused by this choice.
- Regarding uncertainties stemming from the input data for AGR calculation, we chose a strategy similar to Buchwitz et al., (2018) approach for calculating their standard deviations for global AGR. Namely, as shown in Figure 8 *"Error bars of $AGR_{TCCON}$ (orange vertical dashed line) are defined based on the station-wise variability using method similar to Buchwitz et al., (2018). Namely, average standard deviation across TCCON stations (AGR) multiplied on $\sqrt{N_{total}}/N$ factor where $N$ – number of stations used, $N_{total}$ – total number of stations in TCCON analysis."* Due to this, on Figure 8 uncertainties of the years with lower number of TCCON observations from lower number of stations are higher.

– Overall, it seems that there are times in the paper where it is a case of the cart driving the horse. Of course it is important and valuable to understand the growth rates and their differences between different datasets, but really what we want to do is use these to diagnose what in our understanding is incomplete and leads to the differences. That's presumably some combination of surface fluxes and atmospheric processes. It is a subtlety, but at times the emphasis throughout the paper of quantifying growth rates is overstated, and should be rephrased to state their importance in interpreting the underlying biogeochemistry/physics.

Response 3.15: Perhaps, it was also a problem of a slight mismatch between the research aim formulated stated and the contents we present in the first version of the

paper. Please see that we have reformulated the research aim and the objectives as shown below.

- New research aim is "Our study aims to assess the robustness of GR estimates from the observations of Total Carbon Column Observing Network (TCCON) considering the existing spatio-temporal gaps of the network".
- New objectives are "(a) to estimate the robustness of $AGR_{TCCON}$ due to the data sampling, measurement gaps or difference in time series across the sites. Secondly, (b) to examine the $AGR_{TCCON}$ agreement with the existing $CO_2$ growth references and its sensitivity to external factors Thirdly, an additional objective is set (c) to assess the exposure of $MGR_{TCCON}$ estimates at each TCCON station to external factors".

Despite it is critical to diagnose missing processes behind $CO_2$ growth, the main aim of this article is driven by not less important motivation. Namely, as it is stated now in the introduction *"The $CO_2$ growth rate (GR) is a relatively well-known quantity but there are few global observational approaches suitable for quantifying global GR. Our study aims to assess the robustness of GR estimates from the observations of Total Carbon Column Observing Network (TCCON) considering the existing spatio-temporal gaps of the network"*

Minor Comments/Technical Corrections

The article requires careful proof-reading as there are many grammatical errors. We have attempted to report these, but the list is probably not comprehensive.

Response 3.16: However, this version of the manuscript has undergone considerable revision and the language was a subject of the revision as well. We incorporated the language corrections according to the recommendations from the English native speakers from the TCCON community side. For instance, language-related suggestions of Nicholas Deutscher provided in the collective comment and the language corrections by Debra Wunch in the personal email correspondence.

– line 9: remove "the" before "global warming" - OK

–line10-11: "Despite atmospheric CO2 growth rate had been considered as the well-known quantity" This wording doesn't work. Maybe "Despite the atmospheric CO2 growth rate having been considered as a well-known quantity" - Reformulated

– line 13: Please correctly define TCCON as the "Total Carbon Column Observing Network" - OK

– line 23-25: The structure of this sentence is confusing, and forms a double negative. It would perhaps be better to emphasize good agreement at 85% of stations. - Reformulated

– line 27: "perfect (r=0.99)" - not quite perfect – r = 0.99 correlation is denoted as "excellent"

– line 29: in-> a; also again, this is not "perfect". OK

– line 29: insert "a" before "spatial" - Reformulated

– line 33-36: sentence seems back to front. Missing 'a' before CO2 - Reformulated

– line 42: "permanent" - we'd like to hope it isn't permanent, and while perhaps that is optimistic, a better word should be chosen here – "Persistent" instead of "permanent" is probably better choice?

– line 43-44, and other locations: when referring to growth rate (GR) it should almost have "the" before it. E.g. here, it should read "The atmospheric CO2 growth rate (GR)". There are many instances of this throughout the manuscript that we will not explicitly point out each time. – Corrected to "The atmospheric CO2 growth rate" when this term is used in the text.

– line 46: "the precision of direct observations are high (0.09 ppm)" Please add what kind of instrument has such high precision with citations. In situ measurement? Picarro? Also, "high" is ambiguous, and would perhaps be better replaced by something unambiguous ("excellent") or re-wording ("direct observations are highly precise").

Response 3.17. We reformulated it in this way: *"As the direct observations of $CO_2$ by infrared analyzers in the background atmospheric conditions are precise (accuracy > 0.20 ppm), the GR in the entire atmosphere is known with high confidence."* We used (Dlugokencky and Tans, 2018) as a reference and information about 0.20 ppm precision is taken from the latest respective website (https://www.esrl.noaa.gov/gmd/ccgg/about/co2_measurements.html).

– line 47: "uploaded to" - a better phrasing might be "reported in" OK

– line 48: at-> "on a" OK

– line 50: than-> that OK

– line 51: "constrains" can have a specific meaning in flux estimation, perhaps replace with "limits" OK

– line 61: insert "the" before terrestrial OK

– line 67: process-> Processes Reformulated

– line 70: insert comma before which, remove "per se" OK

– line 71: ecosystem-> Ecosystems OK

– line 72: semiarid-> semi-arid - OK

– line 73: "Despite the regions"-> "Despite the fact that the regions" Reformulated

– line 74: evidences-> evidence (and are ->is) Reformulated

line 79: "the limited"-> "a limited" an alternative is to qualify "the limited number of stations at which measurements are available on long time scales" or something similar - Reformulated

– line 79-81: do we really want to know the growth rate everywhere? You probably want a growth rate for each distinct region (ecosystem or whatever). Your desire to know it everywhere perhaps points to a limitation of the study.

Response 3.18: We are interested in the global growth based on aggregated TCCON measurements. Please note that we reformulated the main research aim and objectives of the study accordingly (below). This change has been motivated by multiple reviewers' comments about research aim-results inconsistencies.

- New research aim is "Our study aims to assess the robustness of GR estimates from the observations of Total Carbon Column Observing Network (TCCON) considering the existing spatio-temporal gaps of the network".
- New objectives are "(a) to estimate the robustness of $AGR_{TCCON}$ due to the data sampling, measurement gaps or difference in time series across the sites. Secondly, (b) to examine the $AGR_{TCCON}$ agreement with the existing $CO_2$ growth references and its sensitivity to external factors Thirdly, an additional objective is set (c) to assess the exposure of $MGR_{TCCON}$ estimates at each TCCON station to external factors".

Each site's growth rate can be affected by local factors, that may or may not be of interest for the growth rate, and not on a global scale. By selecting a simple 12-month difference method, you leave yourself vulnerable to transient effects from local signals, as does failing to sub-sample measurements to remove local effects. Of course there is a role for both understanding the regional scale effects and changes in local factors as well, but you need to be clear about what you are trying to achieve here.

Response 3.19:
- The example of the role of local factors influencing global growth rate is shown in section 3.4.2. The promising finding about low sensitivity of global

CO₂ growth to such strong local influence as exposure to large city $CO_2$ emissions indirectly points to stability of the current methodology and the use of simple sub-sampling strategies.

- The influence of data irregularity is thoroughly investigated in this version of the article (see objective (a) of this study and the analysis for Figure 2, upper panel of Figure 3, Table 1, orange area of Figure 7).

– line 85: ratio-> ratios - OK

– line 88: remove semicolon - OK

– line 96: add "the" before "Global Carbon Budget" – Corrected throughout the text

– line 100: "the TCCON"-> "TCCON's" Corrected throughout the text

– line 101: "approve"->"prove" Reformulated

– line 102: "rest"->"remaining" Reformulated

– line 105: "At second"-> "Secondly" (similarly next line for "At third" Corrected throughout the text

– line 108: "Sect.s"->"Sects" OK

– line 113: Spell out Section here, as you are not explicitly referencing a separate section. OK

– line 118: the measurements are not continuous because they rely on the presence of sunlight, so by nature cannot measure at night or in cloudy conditions. Quasi-continuous maybe, or ongoing OK

– line 119: "The XCO2 estimates are retrieved using the ratio" XCO2 is not retrieved. Replace with "The XCO2 values are obtained by taking the ratio" OK

– line 120: O2 (subscript) OK

Response 3.20: Most corrections from the fragment above applied, we did not apply the corrections only to those formulations disappeared or reshaped as a result of the revision.

– line 121-123: You need references here for these statements.

Response 3.21: The reference for several statements is Wunch et al., 2011 work. We mentioned this reference only once in the end of the entire block that belongs to this reference. More specifically, the statement about insensitivity of column-averaged dry-air mole fractions is given in the introduction of Wunch et al., 2011. Lines 11-12 of their work: *"Column-averaged dry-air mole fractions (DMFs; denoted XG for gas G) are particularly useful for this purpose because they are insensitive to variations in surface pressure and atmospheric water vapour"*.

– line 123: "spectra by pointing on the sun in the near-infrared spectrum" "spectra in the near-infrared region by pointing at the sun" OK

– line 123: superscript OK

– line 125: delete "XCO2"; accuracy-> uncertainty (2-sigma) OK

– line 126: "calibrated" is actually the wrong word here, because it is not truly a calibration. They are compared to these measurements, which links them to the WMO scale OK

–line 127: "from World Meteorological Organization onboard" "traceable to the World Meteorological Organization scale" OK

– line 130: exist->existing OK

– line 131: "familiarize"->"familiarize themselves" OK

– line 133: "addresses to"->uses OK

– line 135-137: not clear what this sentence means - needs clarification -Reformulated

– line 145/Figure 1: In this figure, it appears that the green dot "ZUG" should in fact be "KAR" for Karlsruhe. ZUG and GAR are practically co-located and would therefore not be differentiated on this scale map.

Response 3.22: Figure is remade

– line 150: Insert "The" at the start of the sentence, and "is" before "being" OK

– line 151: "developed by the Institute of Pierre Simon Laplace (LSCE)" LSCE stands for "Laboratoire des Sciences du Climat et de l'Environnement" OK

– line 157: Sentence starting here needs clarification/rewording. Reformulated

– line 160: insert "the" before biomass OK

– line 161: insert "the" before "fire module" and needs clarification on GFAS (GFAS emissions possibly, or "uses the GFAS") OK

– line 164-165: how did you judge that "this process did not cause serious error propagation"?

Response 3.26: We omitted the comparison figure from the supplement due to similarity of the models' resolution but can insert it back if needed. This fact is also seen by the similar agreement between TCCON-CT and between TCCON-CAMS where the agreement with CAMS is not deteriorated. Also, by high spatial agreement between CAMS and CT on Figure 13.

– line 175: cover->coverage OK

– line 177: "CTA" What does this stand for? Replace with "CT"? OK

– line 177-179: please be clear that there are multiple modules for each flux within CarbonTracker. E.g. while CASA is used in both, there are two versions of the biosphere model.

Response 3.27: CASA includes GFED 4.1s (hourly resolution) and GFED_CMS (daily resolution)

– line 178: "GFED" is already defined in line 161 OK

– line 188, 221, 224: subscript $_2$ OK

– line 191: "being updated at"->"is updated on" OK

– line 192: why "Obviously"? Suggest removing this word. OK

– line 195: on-> at; insert "The" before GCB OK

– line 196: The correct journal name is Earth System Science Data OK

– line 197-199: Sentence needs to be revised. Either "Large scale AGR estimates... are also taken from" or "Another reference ... is the satellite data" (i.e. delete "are taken from" in this second option) OK

– line 201: Should be "Greenhouse gases Observing SATellite" OK

– line 210: is there also a neutral classification (0)?

Response 3.28: True, the sentence is corrected.

– line 226, line 577: Piao et al., 2019 –> Piao et al., 2020 OK

– line 240: about the definition of AGR, MGR was a difference between 2 successive years. But for AGR, isn't it also a difference between 2 successive years? Clarify the difference

Response 3.29: Please see updated more detailed description of MGR calculation (Equation 3) and AGR calculation (Equation 4).

– line 252: How do you determine the number of observations per month. Some sites routinely make several hundred per day, so presumably some pre-processing is done.

Response 3.30:

- The number of monthly observations is defined based on the availability of mean daily $XCO_2$ estimates from a TCCON station (the details about this procedure are shown in Figure 2 and section 3.1).
- Thanks for pointing out the irregularities in daily data abundance across the TCCON sites. Based on this, the current version of the paper analyzes the role of summarized sub-daily observations performed at each TCCON sites. For details please see Table 2 and the associated description in the paragraph above. The total number of daily observations is for each station is also illustrated on Figure S.2.2 (supplementary).

– line 252: less -> fewer Also, this still seems fairly loose, would it not be better to raise the minimum number of observations contributing to the monthly median? This is discussed later, but there is no reference to that later discussion.

Response 3.31: Raising the minimum daily threshold leads to wiping out many TCCON stations as shown in the section 3.1 and Figure 2. At the same time, the difference between the minimum amount of the sub-monthly input and somehow medium amount is surprisingly low (error spread from Figure 3 and orange area from Figure 7). Due to this, increasing the daily threshold does not lead to improvement between $AGR_{TCCON}$ and the references as we made a note about this in the study:

*"change of daily threshold from '2' to higher values (3-25) does not lead to improved correlation of AGRTCCON with either SAT ($r \leq 0.60$ for any threshold) or GCB ($r \leq 0.56$ for any threshold). At one hand, these findings confirm the correctness of our approach of saving as much data as possible by using daily threshold '2'."* In other words, we have not found any evidences that raising the number of the observations in the minimum threshold would strengthen the analysis and use the '2' threshold.

– line 266: "from 1.71 ppm (2009) to 3.35 ppm (2008)" => "from 1.71 ppm (2010 or beginning of 2011?) to 3.35 ppm (2009 or beginning of 2010?)"

Response 3.32: This sentence is reformulated according to the new results. We checked the consistency of AGR estimates with the years mentioned in the text.

– line 272: do you try selecting the model or satellite data to match the TCCON spatio-temporal measurement pattern? This would confirm or refute your assumption.

Response 3.33: We use satellite global growth estimates from previous study as the global-scale reference (Buchwitz et al., 2018). Another reference is from the global carbon budget (Friedlingstein et al., 2019). Moreover, the hypothesis about the period-driven agreement between $AGR_{TCCON}$ and the references is rejected based on the results from the current version.

– line 273 "a study period"->"the study period" OK

– line 275: r = 0.61 for TCCON-SAT, r = 0.49 for TCCON-GCB In the Abstract and Discussion sections, the correlation coefficients are described as being 0.75 for TCCON-SAT and 0.68 for TCCON-GCB. Which is correct? Reformulated based on new results

– line 278: "right below"->"directly below the" OK

– line 281: not sure why you would expect vegetation-driven seasonality to result in wave-shaped fluctuations. Interannual variability, perhaps, but by the way you are calculating it you are accounting for the seasonal cycles.

Response 3.34: That is true. Seasonal-dependent analysis of MGRs at the selected sites confirms the lack of vegetation-driven seasonality (section 3.2.1). *"We report the estimates of seasonal MGRs in the brackets using the following order: winter, spring, summer and autumn (TSU = 2.57, 2.02, 2.54, 2.70 ppm; GAR = 2.62, 2.19, 2.26, 2.21 ppm; PKF = 2.03, 1.80, 2.24, 2.56 ppm)".*

– line 283: approve? Not the correct word here Reformulated

– line 287: "(right axis of Fig. 1, panel b)"->"(right axis of Fig. 2, panel b)" OK

– line 291: "to data"-> "with data" Reformulated

– line 292: remove comma Reformulated

– line 295: "Besides 15-year pattern" - not sure what you mean? Reformulated

– line 298: "yield to"->"yield a" OK

– line 300: This underscores the importance of comparing apples with apples, basically. That is, if you want to make a comparison between similarities in behaviour either across time periods or between datasets, then they need to be sampled to minimize potential spatio-temporal biases.

Response 3.35:

- To ensure there is no pitfall with "apples" here, we analyzed the role of daily, monthly and annual data to station-wise and global $AGR_{TCCON}$ estimation (Table 1 and the corresponding description above).
- The uncertainty in Figure 8 (what was Figure 2 in the previous version) reflects the data input irregularity driven by varying number of stations used for calculation global $AGR_{TCCON}$ for every year. This is a similar step to Buchwitz et al., (2018) approach of AGR uncertainty calculation where instead of months we use number of stations. As methodology subsection and Figure 8 caption state for instance *"average standard deviation across TCCON stations (AGR) multiplied on $\sqrt{N_{total}}/N$ factor where N – number of stations used, $N_{total}$ – total number of stations in TCCON analysis"*

- We also followed the TCCON reviewer's recommendation from the comment with response 3.38 (below) and analyzed relationship between abundance of MGR for every year and TCCON-to-reference bias (see the response 3.38).
- To be aware which of our stations are more affected by seasonality or time series edge effects, we compared AGR with the six-parameter-based linear trend calculated by Lindqvist et al., 2015 methodology.

– Figure 2 and its caption: the scale used for the lines corresponding the AGR_XCO2 is not exactly every 8th month (August) as mentioned in the caption, for example for the peak of the gray line (AGR_TCCON) which is located around 2nd month of 2010.

– Figure 2 caption - missing subscripts on the AGR terms.

Response 3.36: This figure is corrected

– line 312: "during the year where periods starting from winter months are shown with green tones, from spring with gray tones, from summer with golden tones, from autumn with red tones." Clarify about differences for different hemispheres. For once, this seems to have a southern hemisphere bias->

Response 3.37: This does not seem to be southern hemisphere bias according to new Figure S.2.3 (supplementary material) where we split all MGR to northern and southern hemispheres according to the recommendation above.

– line 319: "be risen"->"arise" OK

– line 320 "(based on one-time TCCON observations)" Is it one measurement of TCCON or is it a daily mean measurement of TCCON? Reformulated

– line 321-322: Not sure what this sentence means - clarify. Reformulated

– line 329: "different thresholds we mentioned above"->"the different thresholds mentioned above" OK

– line 332-335: perhaps it would be better to correlate the number of available MGR with the ratio of TCCON to SAT (or model) error spreads, or again compare by subsampling the model or satellite data to resemble the relevant TCCON data for each period.

Response 3.38: We followed the recommendation and correlated the number of available MGRs (and AGRs) with the ratio of TCCON-to-SAT and TCCON-to-GCB bias for all possible daily thresholds. There is no strong correlation for $MGR_{count}$ vs TCCON-to-$GCB_{bias}$ comparison ($r = 0.22 - 0.41$ depending on daily threshold) and for $MGR_{count}$ vs TCCON-to-$SAT_{bias}$ comparison ($r = 0.25 - 0.46$). For daily threshold

of '2' the correlation coefficients for the aforementioned comparisons are 0.34 (vs TCCON-to-GCB) and 0.44 (to-SAT) respectively.

– line 347: such as 30 points Reformulated

– line 349: approves->proves OK

– line 363: versus->with OK

– line 367: add "the" before "correlation" Reformulated

– line 378: plain->simplistic (assuming this is what you mean) OK

– line 381: regarding the Tsukuba site - probably the topography cannot be resolved at these model resolutions. Also, if the difference between models and TCCON is driven by the land carbon sink as you hypothesize, surely the neighboring sites (Rikubetsu, Saga) should exhibit similar behavior.

Response 3.39: Please see the 3.10 and 3.11 responses as the comments behind these responses have almost identical suggestions.

– line 387: It's not "most likely" that Ascension is a small island-> Please fix this wording. Though as noted in the general comments it is not clear that this small landmass would have affect model-measurement differences here in the regionally-representative column.

Response 3.40: We fixed this awkward wording. Please see the additional analysis of biomass burning fluxes role incorporated at Figure 14 and the corresponding section above. Ascension is analyzed in details in this version of the paper (section 3.4.3 plus supplementary Figure S.2.6).

– line 397: add "themself" after familiarize OK

– line 400/Figure 4: It looks like CAMS is missing for WOL, and CT missing for ASC. It would be helpful to include the values for the coefficients on the plot.

Response 3.41: Please check new version (Figure 12) with correlation coefficients plotted at the edge of the bars.

– line 415: on-> by? OK

Seems fundamentally like the average AGRs should be in good agreement, unless there is something wrong with a dataset, or the comparative data are sampled so as to introduce biases between them. This is again an instance where the models/satellites should be sampled in the same spatio-temporal fashion as TCCON before drawing any conclusions about comparative AGRs.

Response 3.42:

- We use the Global Carbon Budget and satellite data from Buchwitz et al., (2018) as the AGR references representative for global scales of $CO_2$ growth. These datasets are sampled by those researchers who generated the respective datasets and published these results. SAT and GCB are not necessarily sampled in the same way but as they represent the global growth, the agreement between them is high (r = 0.87). Therefore, we expect high agreement between the AGR references and (presumably) globally representative $AGR_{TCCON}$.

- $AGR_{TCCON}$ estimates do exhibit high agreement with AGR references as shown *"$AGR_{TCCON}$ strongly agrees with SAT (r = 0.83) and with GCB (r = 0.82) identically resembling SAT-to-GCB mutual agreement in global AGR reproduction (r = 0.83)"* except years 2008 and 2015.

- The issues from 2008 and 2015 are identified based on AGR-ENSO comparison shown on Figures 10 and 11. These are the years when probably ENSO-driven bias has influenced the accuracy of $TCCON_{AGR}$ estimates. As we stated, *"Despite it is challenging to outline the exact mechanism of this finding, if we assume 2008 and 2015 years were impacted by excessive or irregular sensitivity of some TCCON stations to short-term ENSO conditions."*

- As we are unable to explain the exact mechanism of the TCCON-to-reference bias exposure to ONI, we welcome any suggestions about the phenomenon of the $AGR_{bias}$-to-ONI agreement in MJJ (May-June-July) period (see Figures 10 and 11).

– line 416: From->"From a" This sentence is removed from the current version.

– line 420: this is not perfect correlation.

Response 3.43. Perfect correlation is everywhere reformulated to "excellent" if r = 0.98-0.99

– Figure 5: In the "Global" panel (top right side), there are TCCON data for 2006 and 2007 (gray bars), but there aren't any in the 3 other panels (neither in "NH", nor in "SH" or "Tropical"). Maybe emphasise that there must be a minimum 2(?) sites contributing within each region.

Response 3.44: The regional analysis is removed from the current revision due to lack of valuable information (for new research aim and objectives) for the current analysis and following several recommendations.

– line 434: variability->coverage or representation Reformulated

– line 436: dimensions? Do you mean on different scales? Yes. Reformulated

– line 438: errors-> differences OK

– line 439: it's not clear how the time lag affects model AGR differences, unless there are differences in the models capturing this.

Response 3.45: True. We also did not notice this problem based on our data, so this assumption is not necessary. Sentence is reformulated, we mention only transport-model-driven errors.

– line 446: citations needed here

Response 3.46: To avoid misunderstanding, we reformulated the sentences by using word "reference data from satellite and global carbon budget" so a reader can understand we have in mind the reference data used in this article. The respective references are also added.

– line 454: again, not "perfect" agreement OK

– line 456: Oceans doesn't need to be capitalized OK

– line 474-475: Not sure what this sentence means. Reformulated

– line 476: that->as OK

– line 477-481: lots of reporting of statistics without any need. This could be simplified.

Response 3.47: This part of paragraph is shortened.

– line 492: approves->supports OK

– line 493: not perfect OK

– line 498: "oppositely" - suggest replacing with an alternative word. OK

– line 501: at->in the OK

– line 504: "of by"-> "by" OK

– line 511: again, not perfect OK

– line 515/Table 2: Please add units for "Difference". OK, 'ppm' added

– Table 2 and lines 493-500: The median differences and correlation co-efficients are incosistent between the table and the text.

Response 3.48: Inconsistencies are corrected

– line 530: agreements->agreement OK

– line 538-540: revise as appropriate once earlier section is revisited Reformulated

– line 542: another misuse of "perfect", and in this case I wouldn't even call them near-perfect. Reformulated

– line 552: not perfect OK

– line 553: sentence needs revising ("on > 90%" doesn't make sense)

Response 3.49: In the new version, the formulations in these paragraphs are presented in different way and this reference is not required anymore.

– line 555: not perfect OK

– line 558: 0.02-0.03ppm->0.04 ppm? (c.f. line 494 and 495) OK

– line 566: "The results of this study have three vectors of implications." What does this mean? Why not just say there results have three implications? OK

– Conclusions: as noted earlier, "at second" and "at third" should be replaced by "secondly" and "thirdly" OK

– line 567: approves->confirms; old->existing OK

– supplement, line 8: Table S.2.2-> Table S.2.1 OK

– supplement, line 64, 85: subscript $_2$ OK

References

Lindenmaier, R., Dubey, M. K., Henderson, B. G., Butterfield, Z. T., Herman, J. R., Rahn, T., and Lee, S.-H.: Multiscale observations of CO2, 13CO2, and 265 pollutants at Four Corners for emission verification and attribution, Proceedings of the National Academy of Sciences,111, 8386–8391, https://doi.org/10.1073/pnas.1321883111, http://www.pnas.org/content/111/23/8386.abstract, 2014.

Lindqvist, H., O'Dell, C.W., Basu, S., Boesch, H., Chevallier, F., Deutscher, N., Feng, L., Fisher, B., Hase, F., Inoue, M., Kivi, R., Morino, I.,Palmer, P. I., Parker, R., Schneider, M., Sussmann, R., and Yoshida, Y.: Does GOSAT capture the true seasonal cycle of carbon dioxide?,Atmospheric Chemistry and Physics, 15, 13 023–13 040, https://doi.org/10.5194/acp-15-13023-2015, https://www.atmos-chem-phys.net/270 15/13023/2015/, 2015.

Yuan, Y., Sussmann, R., Rettinger, M., Ries, L., Petermeier, H., and Menzel, A.: Comparison of Continuous In-Situ CO2 Measurements with Co-Located Column-Averaged XCO2 TCCON/Satellite Observations and CarbonTracker Model Over the Zugspitze Region, Remote Sensing, 11, 2981, https://doi.org/10.3390/rs11242981, http://dx.doi.org/10.3390/rs11242981, 2019.

Response 3.50: The suggested references were added to the manuscript.

---

## Author Comment (AC4) · 5 Dec 2020

**RESPONSE TO THE REVIEWER #2**

In their work "The consistency between observations (TCCON, surface measurements and satellites) and $CO_2$ models in reproducing global CO2 growth rate", submitted to ACP, the authors investigate the agreement of the atmospheric CO2 (annual) growth rates from several data sources: total column CO2 from the network of ground-based Fourier Transform Spectrometers (TCCON), inverse model estimates from Carbon-Tracker 2017 and CAMS, and reported growth rates from the Global Carbon Budget as well as total column $CO_2$ from satellites. While quantifying the atmospheric $CO_2$ growth rate using different data sources might be a topic suitable for ACP, the scientific questions that this paper attempts to address are not sufficiently clearly defined. Thus, the research presented in the paper seems to somewhat suffer from this throughout the manuscript. The concepts, ideas, tools and data in the paper are not novel, and the methods used for data analysis are not entirely valid or sufficiently described. I consider that the paper should not be accepted for publication in its current form. A thorough revision starting from re-formulating the research questions would be necessary. In what follows, I will point out more specifically my concerns and comments on the manuscript, focusing more on the early sections as they naturally affect the rest of the manuscript. Because the comments are quite extensive, I will exceptionally not list technical corrections nor specific comments related to the English language in my review.

Response 2.01: We thank the reviewer for the valuable suggestions based on which we have substantially improved the manuscript. Point-by-point responses are given below and marked by green color. Please note that the definition of GR, AGR and MGR terms are given in the article. We have followed the recommendation for thorough revision in the way proposed by the reviewer in the comment above. Foremost, we reformulated the research question and objectives to be more specific (also after many comments from TCCON community) as following.

- New research aim is *"Our study aims to assess the robustness of GR estimates from the observations of the Total Carbon Column Observing Network (TCCON) given the importance of the network to the global carbon monitoring and its expanding observational coverage."*
- New objectives are "*(a) "to estimate the robustness of $AGR_{TCCON}$ due to the data sampling, measurement gaps or difference in time series across the sites". This objective is now supported by expanded evidences about daily, monthly, annual and seasonal stability of TCCON AGR estimates, objective*
- *(b) "to examine the $AGR_{TCCON}$ agreement with the existing $CO_2$ growth references and its sensitivity to external factors".*
- *(c) to assess the exposure of $CO_2$ growth estimates at each TCCON station to external factors*

Specific comments:

Lines 64-89: The main message of this paragraph is unclear to the reader: are we interested in the global or local CO2 growth rate in this paper? The authors start by highlighting the need for an accurate knowledge of the global $CO_2$ growth rate; however, their analysis focuses on both local and global analysis without a clear focus or guidance to what essentially is important in the results and why the analysis has been made.

Response 2.02: According to the reformulated research aim, we are interested in the global $CO_2$ growth reproduced by TCCON global observations. Please note that the introduction was restructured accordingly where the 3rd paragraph is now dedicated to the importance of TCCON-based $CO_2$ growth estimation.

On lines 97-104, the authors state the three objectives of their study. Regarding these objectives (a)-(c), (a) I don't think that aggregating all TCCON data would represent the global $CO_2$ growth rate because of several reasons: even though the TCCON is in principle a (near-)global network, the instruments are mainly located in North America and Europe. In addition, several of them are affected by local sources (e.g., Paris, Tsukuba, Pasadena) that make them interesting for the evaluation of satellite $CO_2$ retrievals in urban circumstances but maybe less representative of the global $CO_2$ background for the purposes of this study.

Response 2.03:

- We think that our study has provided several proofs about relative suitability of TCCON observations for calculating $CO_2$ global growth. Namely: satisfactory stability of $CO_2$ growth estimates despite high variability of data availability from station to station and from year to year, reasonable agreement with the global references, low sensitivity of TCCON-based $CO_2$ growth to the presence of urban sites in the aggregated signal.
- The latter finding about urban areas overlaps with the reviewer's concern about the influence of local sources (see the entire section 3.4.2 dedicated to this issue). The urban influence on interannual TCCON signal is expressed by the agreement rate of MGR estimates between TCCON (can be influenced by local source) and the models (are likely too coarse to capture local-source related $CO_2$ variability). TCCON-to-model correlation coefficient has negative agreement ($r = - 0.73$) with the size of the closest megacity to TCCON station (calculated by MODIS urban pixels, megacity = city $> 1500$ $km^2$). Since it can indicate a potential exposure of TCCON station to urban $CO_2$ emissions, we approximately quantified this exposure by using the distance to the closest megacity. According to this analysis, Paris, Tsukuba, Saga, Pasadena and Karlsruhe are theoretically the most influenced stations ($< 40$ km to megacity). However, the reviewer hypothesis about malignant role of these sites in global AGR signal is not supported since *"The difference between original $AGR_{TCCON}$ and $AGR_{TCCON}$ without "the most urbanized sites" ranged from negligibly low ~0.00 ppm to 0.29 ppm (2017) despite these "most urbanized sites" composed >20% of observational cover of TCCON in 2017."*
- Regarding the spatial limitations of the TCCON network, we have discovered the enhanced sensitivity of $AGR_{TCCON}$ estimates to ENSO late spring-early summer (MJJ) anomalies (Figures 10, 11). This may be an indication of irregular response of TCCON stations to ENSO anomalies where AGR of

some stations is more heavily influenced by these anomalies. All the above-mentioned speculations are added to the current version.

Regarding (b) and (c), the estimation of spatiotemporal inconsistencies between inverse models has already been carried out in a number of studies. It is not clear whether this study brings anything new to the discussion. In particular, I found the analysis regarding (c) a bit rushed and shallow, and lacking important references to earlier studies that consider either these particular models or regions of interest (e.g., Lindqvist et al., 2015; Palmer et al., 2019). The analysis focuses on the atmospheric CO2 growth rate and several times mentions a "permanent" growth of CO2, which may be misleading to readers, considering the seasonal cycle of CO2.

Response 2.04: We agree we did not manage to show that the estimation of spatiotemporal inconsistencies between the inverse models is a core of this study. As mentioned, we dedicated most efforts to the TCCON-related analysis in this version (objectives 'a' and 'b') while the model intercomparison now represents additional analysis (objective 'c').

- As the model analysis was shallow, we provided more details about spatio-temporal analysis between $CO_2$ models (agreement between the models and biomass burning, biosphere, fossil fuel and oceanic fluxes plus agreement between the models and the surface $CO_2$ data indicating how well the surface-governed growth is captured by models).
- The word "permanent" is avoided in this revision.
- Lindqvist et al., 2015 and Palmer et al., 2019 are added as references to this study.
- Moreover, Hannakaisa Lindqvist is included as one of the key coauthors in this revision.

Section 2.1.1 (TCCON): The TCCON data policy requires that the authors contact the TCCON PI's in the preparation phase of the paper in order to guarantee that the data are used and interpreted correctly and also to agree on a potential co-authorship in case the TCCON data have a central role in the manuscript. Since the TCCON PI's have already commented on this issue separately, I do not focus on this more. I do want to add, however, that several issues in the manuscript regarding the interpretation of the results at specific TCCON sites would have been clarified in the preparation process of the manuscript in case the TCCON PIs had been contacted for the work.

Response 2.05: We apologize for the oversight with the TCCON policy. We hope that the new version of the manuscript composed in closer coordination with the TCCON community with their comments applied (see the open access TCCON collective comment at the ACP page of our paper) is significantly improved.

Sections 2.1.2, 2.1.3, and 2.1.4: These sections lack plenty of relevant details, such as version numbers of several of the prior flux components and a more detailed description

of the satellite data, even though these data were cited ("CO2 observations from SCIAMACHY and GOSAT" is not sufficient).

Response 2.06: Sufficient details about all datasets have been added, please see the updated descriptions in the section 2.1 highlighted by yellow color.

Section 2.2.1: Description of the methodology is not sufficient for reproducing the results. For example, it is not clearly described how the gaps in the data are considered.

Response 2.07: Description of the methodology is complemented by additional information (Section 2.1.1) highlighted by yellow color. Since the data used for the AGR estimation as input considerably varies in temporal (daily, monthly, annual) and spatial (station-wise) scales, we devoted one objective of the study to investigating the role of input data characteristics in $AGR_{TCCON}$ estimation. Namely, *"to estimate the robustness of AGRTCCON due to the data sampling, measurement gaps or difference in time series across the sites."* The results from the data structure analysis are shown in sections 3.1 and 3.2.2. The most important expression of the data gaps in the final $AGR_{TCCON}$ estimates is described in the methodology now. They are calculated *"based on the station-wise variability using method similar to Buchwitz et al., (2018). Namely, average standard deviation across TCCON stations (AGR) multiplied on $\sqrt{N_{total}}/N$ factor where N – number of stations used, $N_{total}$ – total number of stations in TCCON analysis"*.

Are there any criteria for including or excluding some of the TCCON sites?

Response 2.08:

- See section 3.1 where we tried to include as much stations as possible using the minimum daily threshold of '2' (please see the same section for explanation of this term).
- Note that to calculate $CO_2$ growth annual growth projected for monthly scales (MGR) for instance, of January 2015, one needs to have not only $XCO_2$ from January 2015, but also from January 2014. Since paired monthly estimates are required, we had to filter out many useful monthly estimates.
- As we have stated, *"there are not enough measurements to calculate a single MGR from MAN and IND stations regardless to the daily threshold (see Figure 2). At several other stations (AMY, BUR, JPL) there is insufficient number of MGRs (< 8) during the entire study period to calculate a single $AGR_{TCCON}$. Hence, 6 stations (FCO, MAN, IND, AMY, BUT and JPL) are not used in this study"*. Here '8' is an arbitrary value that indirectly ensures that sites with too few MGRs to calculate even a single AGR are not used in the final analysis.
- Following the recommendation from one of the TCCON PIs, we have tested the role of AMY missing site in the global AGR (AMY has 3 MGRs at the softest daily threshold of '2'). The impact of these scarce MGR estimates on global AGR would be just within instrumental uncertainties (~0.1 ppm) for 2016 and 2017 years.

Exact methodologies should also be described for comparisons of model and TCCON data (e.g., spatiotemporal interpolation of the gridded model data, averaging kernel corrections etc.).

Response 2.09: We moved section the description about calculation of pressure-weighted $XCO_2$ using model simulations to the main body of the manuscript. See Equations 1, 2, please and the respective description. Simple description is also added about horizontal collocation of TCCON observation and grid cell of the model.

The results and discussion sections suffer from a very scattered analysis which is rich in details but not in content, and lacks focus. Correlation analysis is not sufficient in case of time series: for example, a phase difference in the time series would result in a relatively weak correlation but the reason for the weak correlation would not be identified. At least some representative cases of the $XCO_2$ time series should be presented.

Response 2.10: We tried to enrich the current revision results by

- Including few representative TCCON sites (Tsukuba, Park Falls, Garmisch) for $XCO_2$ vs MGR vs AGR detailed analysis (Section 3.2.1).
- Including additional analysis for Ascension measurement site (Figure S.2.6) where the modeled MGR values are seen as well.
- Regarding the AGR correlation analysis, when we use it for validating $AGR_{TCCON}$ against AGR references, we do not a phase difference between two different estimates of annual growth. We agree this can be the case for AGR-to-ENSO comparison, so the AGR-to-ENSO strength correlation analysis is provided with ± 1-year lag (Figure 9). Moreover, as the AGR-to-ENSO agreement can be driven by some seasonal components of ENSO (expressed as ONI index), we performed analysis of AGR versus every type of ONI (covering all possible 3-month periods during the year) as shown on Figure 10.

The discussions and conclusions drawn on the claimed "biomass burning regions" seem particularly rushed and would have been relatively straightforward to check by the authors because at least CarbonTracker 2017 provides the imposed fire fluxes as a separate data field.

Response 2.11: We incorporated the analysis of Carbon Tracker 2017 fire fluxes (as well as all other components). See the details at Figure 14 and the respective section please.

---

## Author Comment (AC1)

**RESPONSE TO THE TCCON POLICY**

Let me apologize for the oversight with the TCCON policy. The compliance with the TCCON policy has been alleviated as during the revision we had been contacted by the TCCON co-chair, Asia/Oceania (Nicholas Deutscher). Under his assistance, we followed the TCCON Data License and policies and shared the material of the manuscript with all PIs of the TCCON network and those who were interested, provided their interactive comments. We have also added the missing references required by the TCCON policies to the current version. Moreover, we have offered coauthorship to the TCCON researchers responsible for composing TCCON collective feedback. However, so far, all the TCCON researchers decided to keep providing comments without becoming coauthor for this manuscript.

For more detailed information about tailored amendments to the TCCON community comments, see our response to the TCCON comments, please.